# Guanylate-binding protein 1 acts as a pro-viral factor for the life cycle of hepatitis C virus

**Daniela Bender[1], Alexandra Koulouri[1], Xingjian Wen[1], Mirco Glitscher[1], Anja Schollmeier[1], Liliana Fernandes da Costa[1], Robin Oliver Murra[1], Gert Paul Carra[1], Vanessa Haberger[1], Gerrit J. K. Praefcke[2], Eberhard Hildt[1]***

**1** Paul-Ehrlich-Institut, Department of Virology, Langen, Germany, **2** Paul-Ehrlich-Institut, Department Haematology and Transfusion Medicine, Langen, Germany

* eberhard.hildt@pei.de

## Abstract

Viral infections trigger the expression of interferons (IFNs) and interferon stimulated genes (ISGs), which are crucial to modulate an antiviral response. The human guanylate binding protein 1 (GBP1) is an ISG and exhibits antiviral activity against several viruses. In a previous study, GBP1 was described to impair replication of the hepatitis C virus (HCV). However, the impact of GBP1 on the HCV life cycle is still enigmatic. To monitor the expression and subcellular distribution of GBP1 and HCV we performed qPCR, Western blot, CLSM and STED microscopy, virus titration and reporter gene assays. In contrast to previous reports, we observed that HCV induces the expression of GBP1. Further, to induce GBP1 expression, the cells were stimulated with IFNγ. GBP1 modulation was achieved either by overexpression of GBP1-Wt or by siRNA-mediated knockdown. Silencing of GBP1 impaired the release of viral particles and resulted in intracellular HCV core accumulation, while overexpression of GBP1 favored viral replication and release. CLSM and STED analyses revealed a vesicular distribution of GBP1 in the perinuclear region. Here, it colocalizes with HCV core around lipid droplets, where it acts as assembly platform and thereby favors HCV morphogenesis and release. Collectively, our results identify an unprecedented function of GBP1 as a pro-viral factor. As such, it is essential for viral assembly and release acting through tethering factors involved in HCV morphogenesis onto the surface of lipid droplets.

## Author summary

Morphogenesis and release of the hepatitis C virus (HCV) are still not fully understood. Here we describe a novel so far unprecedented pro-viral function of the interferon stimulated gene (ISG) human guanylate binding protein 1 (GBP1). In contrast to the supposed antiviral effect, we observe that GBP1 favors morphogenesis and release of HCV by acting as an assembly platform on the surface of lipid droplets. This extends our knowledge about the functions of GBP1 and the life cycle of HCV and should be considered for curative approaches targeting innate immunity.

**Data Availability Statement:** All data are fully available in a public repository (Mendeley Data)

(doi: 10.17632/jpw686w9x2.1) https://data.
mendeley.com/datasets/jpw686w9x2/1.

**Funding:** This work is supported by the LOEWE center DRUID (D2 to EH) (Novel Drug Targets against Poverty-Related and Neglected Tropical Infectious Diseases) (https://www.loewe-druid.de/). This publication was supported by the European Virus Archive goes Global (EVAg)project, which has received funding from the European Union's Horizon 2020 research and innovation program under grant agreement 653316. Based on this funding, EVAg provided the ZIKV French Polynesia H/PF/2013. The funders had no role in study design, data collection and analysis, decision to publish, or preparation of the manuscript.

**Competing interests:** The authors have declared that no competing interests exist.

## Introduction

At present, an estimated 58 million people worldwide suffer from a chronic infection with the hepatitis C virus (HCV), which is designated as being the major cause of end-stage liver diseases [1]. As a member of the *Flaviviridae* family, HCV is a positive-sensed, single-stranded RNA virus with a 9.6 kb genome that encodes for a single polyprotein precursor of 3100 amino acids [2,3]. Viral replication occurs in replication complexes (RCs), which are in close association with intracellular membranes at the *membranous web* (MW) [4–6]. The assembly of viral particles occurs on the surface of lipid droplets (LDs) in close proximity to ER-derived membranes. After budding of the nascent particles into the ER-lumen, the particles fuse to lipoproteins and are released via different routes that are still under debate [7–9]. Of note, the life cycle of HCV is tightly associated with an increased intracellular reactive oxygen species (ROS) level, which is relevant for the induction of autophagy [10,11]. In line with this, autophagy-related structures play an essential role for the intracellular trafficking, release or degradation of HCV particles [7,11].

Infection with HCV induces the production of type I and III interferons (IFN) that upregulate the expression of interferon-stimulated genes (ISGs) [12]. Among these, the Myxoma resistance proteins (MxA and MxB), the immunity-related GTPases (IRGs), the guanylate binding proteins (GBPs) and the very large IFN-inducible GTPases (VLIGs) present the most prominent ISGs that belong to the IFN-inducible dynamin superfamily [13]. In humans, seven hGBPs have been identified (GBP1-7) with GBP1 being best characterized in terms of links towards antiviral activities. The proximal GBP1 promoter harbors an interferon-stimulated response element (ISRE) and a gamma-activated site (GAS), thus allowing its efficient expression upon stimulation with type I (IFNα, IFNβ) and type II (IFNγ) IFNs [14,15].

GBP1 is a 67 kDa GTPase that hydrolyses GTP to GDP and GMP, with the latter being the predominant product [16,17]. The C-terminal part of GPB1 undergoes post-translational farnesylation at a CaaX-motif allowing its association with intracellular membranes [18].

Initially, GBPs have been identified to function in the defense against bacterial and protozoan pathogens [19,20]. In addition, previous studies indicate that GBPs represent restriction factors that exert antiviral activities against various viruses, as reviewed in [20–22]. For example, GBP1 has been described to interfere with the nuclear trafficking of Kaposi's sarcoma-associated herpesvirus (KSHV) particles and pseudorabies virus by disrupting actin filaments [23,24]. Further, it is involved in targeting HEV capsids towards lysosomal degradation [25]. In case of Influenza A virus (IAV), GBP1 antagonizes the virulence factor NS1 [26]. For HCV it was described that the expression of GBP1 is decreased, which was based on data originating from a subgenomic replicon system. Overexpression of GBP1 exerted an inhibitory effect on HCV replication in this system [27]. Further analyses from the same group revealed that the antiviral effect of GBP1, which is supposed to be based on the GTPase activity, can be counteracted by an interaction between the viral polymerase NS5B and the N-terminal GTPase domain of GBP1 [27,28]. However, the detailed function and mechanism of GBP1 during the HCV life-cycle are not fully understood still.

In contrast to the reports ascribing an antiviral activity to GBP1, a pro-viral effect of GBP1 on Japanese encephalitis virus (JEV), by inhibiting the innate immune response, has been reported [29]. In addition, murine GBP4 and hGBP7 have been identified to behave pro-virally by interfering with factors involved in the antiviral innate immunity [30,31].

Here, we investigated the impact of HCV on GBP1, the role of GBP1 during HCV infection and its potential relevance for the antiviral effect of IFNs. Surprisingly, silencing of GBP1 impaired the release of HCV particles and resulted in an intracellular accumulation of HCV core. In line with this, upon GBP1 silencing, we observed a changed morphology of

LAMP2-positive structures going along with an impaired lysosomal acidification. Otherwise, overexpression of GBP1 favors HCV release and replication. Moreover, in GBP1 overexpressing cells, a vesicular distribution of GBP1 was found. Here, it colocalized with HCV core around lipid droplets as part of the MW, which is involved in the release of viral particles. This further supports its pro-viral role during viral morphogenesis and in the release of infectious HCV particles.

Altogether, our data indicate that, in contrast to the antiviral effects ascribed to GBP1, the host-factor has the capacity to modulate viral assembly and release by tethering of factors involved in HCV morphogenesis.

## Material and methods

### Plasmids

Plasmids encoding HCV genomes pFK-JFH1/GND (replication deficient) and pFK-JFH1/J6 have been described previously [32–34]. pFK-Luc-Jc1 was kindly provided by R. Bartenschlager (Universität Heidelberg, Germany) and has been described by Koutsoudakis et al. [35]. The plasmid encoding pcDNA-Core (gt 1b) has been previously generated in our lab. Plasmids encoding Flag-tagged human GBP1 (GBP1-Wt), pCMV-2B (empty vector) and mCherry-hGBP1 were kindly provided by Gerrit Praefcke (Paul-Ehrlich-Institut, Germany). pGL3-GBP1-Luc was kindly provided from Prof. Ming Li (Second Affiliated Hospital of Soochow University, Jiangsu, China) and has been described by Li et al. [36].

### Antibodies and reagents

The following primary antibodies were used: anti-GBP1 (polyclonal rabbit, 15303-1-AP; Proteintech), anti-core (C7-50) (monoclonal mouse, MA1-080; Thermo Scientific), anti-NS3 (clone 2E3, monoclonal mouse, BioFront Technologies, Tallahassee, USA), anti-LAMP2/CD107B (polyclonal goat, AF6228; BD Biosciences), anti-p62/SQSTM1 (C-terminus) (polyclonal guinea pig, GP62-C; Progen) anti-GAPDH (6C5) (monoclonal mouse, sc-32233; Santa Cruz) and anti-β-actin (AC-74) (monoclonal mouse, A5316; Sigma-Aldrich). For detection of HCV NS5A, a polyclonal rabbit-derived serum was used [37]. Anti-ZIKV NS1 (monoclonal mouse, BioFront Technologies, Tallahassee, USA). For Western blotting, IRDye 800CW and IRDye 680RD were used as secondary antibodies (Li-COR Biosciences). For immunofluorescence staining, Alexa Fluor 488/546/594/633-coupled donkey α-m/r/g secondary antibodies (Invitrogen, Carlsbad, CA) were used as secondary antibodies. For STED experiments, goat anti-rabbit Atto-594 (1:200, 77671-1ML-F, Sigma-Aldrich, Germany), Abberior STAR 635P (1:200, ST635P-1001, Abberior, Göttingen, Germany) and CF568 (1:100, Sigma-Aldrich, Germany) were used as secondary antibodies. Nuclei were stained using 4´,6-diamidino-2-phenylindole (DAPI) (Carl Roth, Germany). BODIPY 495/503 (Thermo Fisher, Germany) and Nile Red (Thermo Fisher, Germany) were used to stain lipid droplets. LysoTracker Deep Red (Thermo Fisher, Germany) was used to stain acidic lysosomes. Phalloidin-Atto 633 (68825-10NMOL; Sigma-Aldrich, Germany) was used to stain F-actin.

### Cell culture and treatments

The highly HCV-permissive human hepatoma-derived cell line Huh7.5 was used for electroporation with HCV RNA. The HCV genomic RNA was synthesized by *in vitro* run-off transcription (IVT) of linearized plasmid DNA (pFK-JFH1/J6, pFK/JFH1-GND) using the T7 Scribe Standard RNA IVT Kit (Biozym, Germany) according to the manufacturer's protocol. Electroporation of the *in vitro* transcribed RNA was performed as described previously [11].

Huh7.5 cells and cells stably replicating HCV were cultivated in Dulbecco's modified Eagle's medium (DMEM; Biowest) supplemented with 10% fetal calf serum, 2 mM L-glutamine, 100 U/ml penicillin, and 100 μg/ml streptomycin (DEMEM complete) at 37°C under 5% $CO_2$ and 90% humidity [38]. Passaging of adherent cells was performed by trypsinization two to three times a week. Stably HCV replicating cells were cultivated not longer than passage six. A549 and Huh7 cells were cultivated under the same conditions. The HCV subgenomic cell line Huh9-13 (Huh7 I377/NS3-3'/wt/9-13) was cultivated in DMEM complete supplemented with 1 mg/ml G418 (Geneticin) that was absent during the experiments. These cells stably replicate HCV but lack the structural proteins, thus not producing infectious particles [39].

HCV stocks for infection experiments were prepared from the stably HCV replicating cells. For ZIKV infection experiments, the ZIKV strain French Polynesia H/PF/2013 was used. The strain was provided by the European Virus Archive goes Global (EVAg).

Treatments with IFNα, IFNβ and IFNγ (Preprotech, Hamburg, Germany) were performed for 24 h at a concentration of 100 U/ml, unless stated otherwise. Leupeptin (Sigma-Aldrich, Hamburg, Germany) was used for 24 h at a concentration of 200 μM. To induce autophagy, the cells were treated with Rapamycin for 24 h at a concentration of 100 nM.

## Transient transfections

For transient GBP1 overexpression, cells were transfected using FuGENE HD Transfection Reagent (Promega, Walldorf, Germany) according to the manufacturer´s protocol or by using linear polyethyleneimine (PEI 25K) (Polysciences, Inc.). Briefly, the cells were seeded at 300.000 cells/well into 6 well plates 24 h prior to transfection. For transfection with PEI, a DNA:PEI mass ratio of 1:6 was used. In case of FuGENE a mass ratio of DNA:FuGENE of 1:4 was used. 16 h post transfection, the medium was exchanged and cells were cultivated for another 48 h.

Knockdown experiments were performed using siPORT *NeoFX* Transfection Agent (Invitrogen, Carlsbad, USA) according to the manufacturer´s protocol using 2 nM GBP1 siRNA [40] or scrambled siRNA (sc-37007, Santa Cruz Biotechnology, Heidelberg, Germany) as a control. The siRNA transfections were performed in 12 well plates by simultaneously transfecting and plating the cells (reverse transfection). For one well, 0.2 μl siRNA (10 μM) (or scramble siRNA (10μM)) were diluted with 49.8 μl Opti-MEM I (Thermo Fisher) and 3 μl siPORT siPORT *NeoFX* Transfection Agent were mixed with 47 μl Opti-MEM I (Thermo Fisher). The two solutions were mixed, incubated for 15 min at room temperature, and the RNA/ siPORT *NeoFX* Transfection Agent complexes were dispensed into the wells of a 12 well plate. The solutions were topped with 900 μl of a cell suspension containing 10.000 cells in DMEM complete, gently mixed and incubated for 72 h without changing the medium. IFNγ treatments (100 U/ml) were performed without media exchange 48 h post transfection.

## RT-qPCR

Total RNA was isolated using the RNA-Solv Reagent (Omega Bio-Tek, Georgia, USA) according to the manufacturer´s protocol. Equal amounts (5 μg) of total RNA were reverse transcribed into cDNA using RevertAid H Minus Reverse Transcriptase (Thermo Scientific) as described previously [11]. The cDNA was diluted 1:10 and used for qPCR analyses using specific primers and the 2x Maxima Probe SYBR Green/ROX qPCR Master Mix (2x) (Thermo Scientific). Data were analyzed using the ΔΔCT method and normalized to the amount of RPL27 transcripts. The following primers were used: GBP1-fwd (5´-ggt cca gtt gct gaa aga gc-3´), GBP1-rev (5´- tga cag gaa ggc tct ggt ct -3´) [41], GBP2-fwd (5´- cta tct gca att acg cag cct-3´), GBP2-rev (5´-tgt tct ggc ttc ttg gga tga-3´), GBP5-fwd (5´- cca tgt gcc tca tcg aga act -3´),

GBP5-rev (5´- aca ggt tgc gta atg gca gac -3´) [29], JFH1-fwd (5'-atg acc aca agg cct ttc g-3'), JFH1-rev (5'-cgg gag agc cat agt gg-3'), hRPL27-fwd (5'-aaa gct gtc atc gtg aag aac-3') and hRPL27-rev (5'-gct gct act ttg cgg ggg tag-3').

To quantify extracellular HCV genomes, viral RNA was isolated from cell culture supernatants via the QIAmp Viral RNA Mini Kit (Qiagen). Extracellular HCV RNA was monitored using LightMix Modular Hepatitis C Virus Kit (53-0557-96; TIB MolBiol, Berlin, Germany) in combination with LightCycler Multiplex RNA Virus Master (6754155001; Roche Diagnostics, Mannheim, Germany) according to the manufacturer's instructions.

## Luciferase reporter assay

Huh7.5 cells or HCV-replicating Huh7.5 cells were transiently transfected with pGL3-GBP1-Luc plasmid DNA (encoding for a GBP1 promotor-driven firefly luciferase) [36]. After 48 h, cells were lysed by adding luciferase lysis buffer (25 mM Tris-HCl, 0.1% Triton-X 100, 2 mM DTT, 2 mM EGTA, 10% glycerol, pH 7.5). The luciferase activity of lysates was determined by automated addition of 50 μl luciferase substrate (25 mM Tris-HCl pH 7.8, 5 mM MgCl2, 33.3 mM DTT, 0.1 mM EDTA, 470 μM Luciferin, 530 μM ATP) and subsequent detection of the chemiluminescence by an Orion II microplate Luminometer. All experiments were carried out in triplicates. Luciferase activity was normalized to the total protein amount of the corresponding lysate as determined by Bradford Assay.

To determine HCV replication, Huh7.5 cell were electroporated with the bicistronic replicon pFK-Luc-Jc1 and analyzed after 72 h as described above.

## SDS-PAGE and western blot analysis

Cell lysates were prepared using RIPA buffer (50 mM Tris-HCl pH 7.2, 150 mM NaCl, 0.1% SDS (w/v), 1% sodium deoxycholate (w/v), 1% Triton X-100) containing protease inhibitors, which was followed by sonication. Equal protein amounts (50–75 μg) were diluted in 1x SDS loading dye, denatured for 10 min at 95˚C and separated by SDS-PAGE containing 12% or 14% v/v acrylamide. Subsequently, proteins were transferred to a polyvinylidene difluoride (PVDF) membrane. The membrane was blocked with 1x ROTI Block solution, followed by incubation with specific antibodies (GBP1-, NS3-, core, GAPDH-, and beta-actin). For detection of specific IgG, the LI-COR CLx Odyssey Imaging System (LI-COR Biosciences, Bad Homburg, Germany) was used. Densitometric analyses were performed using the LI-COR Image Studio Lite v5.2.5.

## Immunofluorescence analysis

Cells were grown on glass coverslips (1.5H) (Carl Roth, Germany) and fixed with a 1:1 mixture of ice-cold ethanol/acetone for 10 min, unless stated otherwise. Primary and secondary antibodies were incubated for 1 h each in a humid chamber at room temperature. For all steps, 5% BSA in TBS-T was used as diluent. After washing with TBS-T, stained cells were mounted onto glass slides with Mowiol 40–88 (Sigma-Aldrich, Hamburg, Germany). LysoTracker Deep Red staining was performed for 1 h at 37˚C prior to fixation with 4% formaldehyde (FA) in PBS for 20 min at room temperature. To visualize lipid droplets, BODIPY 495/503 (Thermo Fisher, Germany) or Nile Red (Thermo Fisher, Germany) were incubated together with the secondary antibodies in cells fixed with 4% FA. Images were analyzed using a Leica Stellaris 8 System (Leica, Wetzlar, Germany) equipped with a white light laser (WLL) for fluorescence excitation and three STED laser for fluorescence inhibition using a 93x objective (HC PL APO 93x/ 1.3 Glyc STED WHITE), 100x objective (HP PL Apo 100x/ 1.4 oil STED WHITE) or an 40x objective (HC PL APO 40x/ 1.1 W CORR CS2). The pinhole was set to 1.0 AU (airy units).

For z-stacks, step sizes were set to 0.2 μm. Dual color STED image acquisition using a single STED line (775 nm) was performed using Atto-594 (1:200) and Abberior STAR 635P (1:200) labeled secondary antibodies, excluding DAPI staining. All images were taken with a bit depth of 12 Mbit. Image acquisition and analysis was performed using the LAS X Software (Leica, Wetzlar, Germany) or FIJI [42]. All immunofluorescence images were deconvoluted via the LasX Lightning Tool using the Adaptive algorithm. Total fluorescence per cell was calculated using FIJI and the following formula: corrected total cell fluorescence (CTCF) = integrated density (area of selected cell × mean fluorescence of background readings). In total, a minimum of ten cells were measured.

### Immunofluorescence stain of liver sections

Formalin-fixed and paraffin-embedded liver sections, gained from HCV seropositive patients and healthy control livers, were coated on slides and deparaffinized for 15 min in xylene. This was followed by rehydration for 10 min in 99% ethanol, 10 min in 75% ethanol and 5 min in ddH$_2$O. Antigen retrieval was achieved by briefly boiling of the slides in 10 mM citrate buffer (pH 6) and incubation for 30 min. The slides were incubated for 60 min in 10% BSA in TBS with 0.1% Tween-20 in a humidity chamber. Primary and secondary antibodies were incubated for 1 h each in a humidity chamber at room temperature. For all steps, 10% BSA in TBS with 0.1% Tween-20 was used as diluent. After washing with TBST the stained cells were mounted onto glass slides with Mowiol (Sigma-Aldrich, Hamburg, Germany). Images were analyzed using a Leica Stellaris 8 System (Leica, Wetzlar, Germany). Image acquisition and analysis was performed using the LAS X Software (Leica, Wetzlar, Germany).

### Virus titration

Virus titers were analyzed based on limited dilution by determining the half-maximal tissue culture infectious dose (TCID$_{50}$) as described previously [43]. Therefore, Huh7.5 cells were infected using serial dilutions of cell culture supernatant (5 steps, 1:10 ratio) in 6 replicates for 72 h. Fixation was performed with ice-cold methanol for at least 30 minutes or overnight at -20°C. Blocking was performed according to the immunofluorescence analysis. For detection of HCV-positive cells, an NS5A-specific antiserum was used [37]. Incubation with the primary antibody was performed overnight at 4°C. Horseradish peroxidase-coupled donkey-α-rabbit IgG (NA934; GE Healthcare, Chicago, IL) served as secondary antibody and subsequent staining was performed using 3-amino-9-ethylcarbazol (30 mM Na-acetate, 12 mM acetic acid, 0.05% w/v 3-amino-9-ethylcarbazol, 0.01% H$_2$O$_2$).

For intracellular TCID$_{50}$, cells were washed with PBS, trypsinized and pelleted. The pellet was resuspended in 1 ml DMEM complete and cells were lysed by three freeze/thaw cycles in liquid nitrogen and a 37°C water bath, respectively. The lysate was centrifuged for 10 min at 13.000 xg and the virus containing supernatant was used for TCID$_{50}$ analyses.

### Statistical analyses

Results are presented as means ± standard error of the mean (SEM) from at least three independent experiments, unless stated otherwise. Statistical analyses were performed using GraphPad Prism 9 (GraphPad Software, CA, USA). For statistical comparisons, normality tests were performed for each data set using Shapiro-Wilk proceeding. For normally distributed data, the two-tailed Student's t-test was used. In case of a non-Gaussian distribution, the data were analyzed using the Mann-Whitney test. For the testing for differences in the means of three or more groups, one-way ANOVA analyses were performed. Statistical significance is represented in figures as follows: *, $p < 0.05$, **, $p < 0.01$ and ***, $p < 0.001$, ns = not significant.

## Results

### Increased amount of GBP1 in HCV replicating cells

Infection with HCV activates the host innate antiviral response resulting in the induction of IFNs and interferon stimulated genes (ISGs). However, HCV has evolved multiple strategies to counteract the IFN response e.g. by interfering with genes of the Janus Kinase (JAK)—signal transducer and activator of transcription (STAT) signaling pathway or with PKR [44,12]. To investigate the impact of HCV on the expression of GBP1, the number of GBP1-specific transcripts was determined by qPCR in HCV positive (Jc1) and the corresponding HCV negative control cells (GND). Here, the expression of GBP1 was significantly increased (~7-fold) by HCV (**Fig 1A**). In accordance to this, reporter gene assays, using a luciferase reporter gene under the control of the GBP1 promoter, revealed an increased luciferase activity in Jc1 cells (**Fig 1B**). Independent from this, an elevated level of GBP1 could be further confirmed by quantitative CLSM on the single cell level (~5-fold) (**Fig 1C**) and by Western Blot analysis (~3-fold) (**Fig 1D**). Notably, Huh7.5 cells contain a mutation in RIG-I and lack a type I IFN response [45]. To rule out the presence of artifacts caused by this, we further compared the GBP1 expression after infection in Huh7 and Huh7.5 cells. However, HCV infection resulted in increased GBP1 levels (~2-fold) still (**Fig 1E–1H**). *In vivo* data corroborated these findings. Specifically, quantitative immunofluorescent analysis of liver sections derived from a patient suffering from chronic HCV, showed a 4-fold higher amount of GBP1 in HCV positive tissue (**Fig 1I**). Interestingly, CLSM- and qPCR-analyses of ZIKV-infected A549 cells, another member of the *Flaviviridae* family, similarly revealed increased GBP1 levels (**Fig 1J and 1K**).

Taken together, these data indicate that GBP1 expression and protein amount are significantly elevated in HCV-positive cells/tissues as compared to the HCV negative cells/tissues.

### The IFNγ-dependent induction of the GBP1 protein is impaired in HCV positive cells

The expression of GBP1 can be induced by IFNs, as the promotor region of GBP1 harbors both an interferon-stimulated response element (ISRE) and a gamma-activated site (GAS). In accordance to this, the strongest induction of GBP1 could be achieved after IFNγ-stimulation of Huh7.5 cells and Huh7 cells, as evidenced by reporter gene assays (**Figs 2A and S1**). Consequently, the following experiments were performed using IFNγ as stimulant. To investigate the effect of HCV on the IFNγ-dependent induction of GBP1, HCV-positive and HCV-negative cells were stimulated with either IFNγ or left untreated (UT) as control. Both reporter gene assays, using a luciferase reporter gene under the control of the GBP1 promoter, and qPCR analyses revealed a reduced IFNγ-dependent induction of GBP1 in HCV-positive cells referred to the respective HCV-negative cells (**Fig 2B and 2C**). However, this reduction is not significant. In contrast to this, single-cell analyses via calculating the corrected total cell fluorescence (CTCF) revealed that HCV significantly attenuates the IFNγ-dependent induction of GBP1 (**Fig 2D**). This might reflect that, in contrast to the other methods, a focus is set on HCV-positive cells in this read-out, which opposes other methods considering a pool of HCV-positive and -negative cells. Regardless of this, IFNγ strongly increases the GBP1 levels in both cases. Interestingly, subcellular analysis via CLSM indicated a changed GBP1 distribution in the HCV-positive cells (Jc1) as compared to the HCV-negative cells (GND) (**Fig 2E**).

In a previous study, IFNγ-dependent GBP1 induction has been described to remodel actin filaments [46]. To exclude a possible effect of the IFNγ-treatment on the actin cytoskeleton in the HCV-positive (Jc1) and HCV-negative (GND) Huh7.5 cells, we performed CLSM analyses. Indeed, we found GBP1 and HCV core colocalizing with actin filaments in the untreated

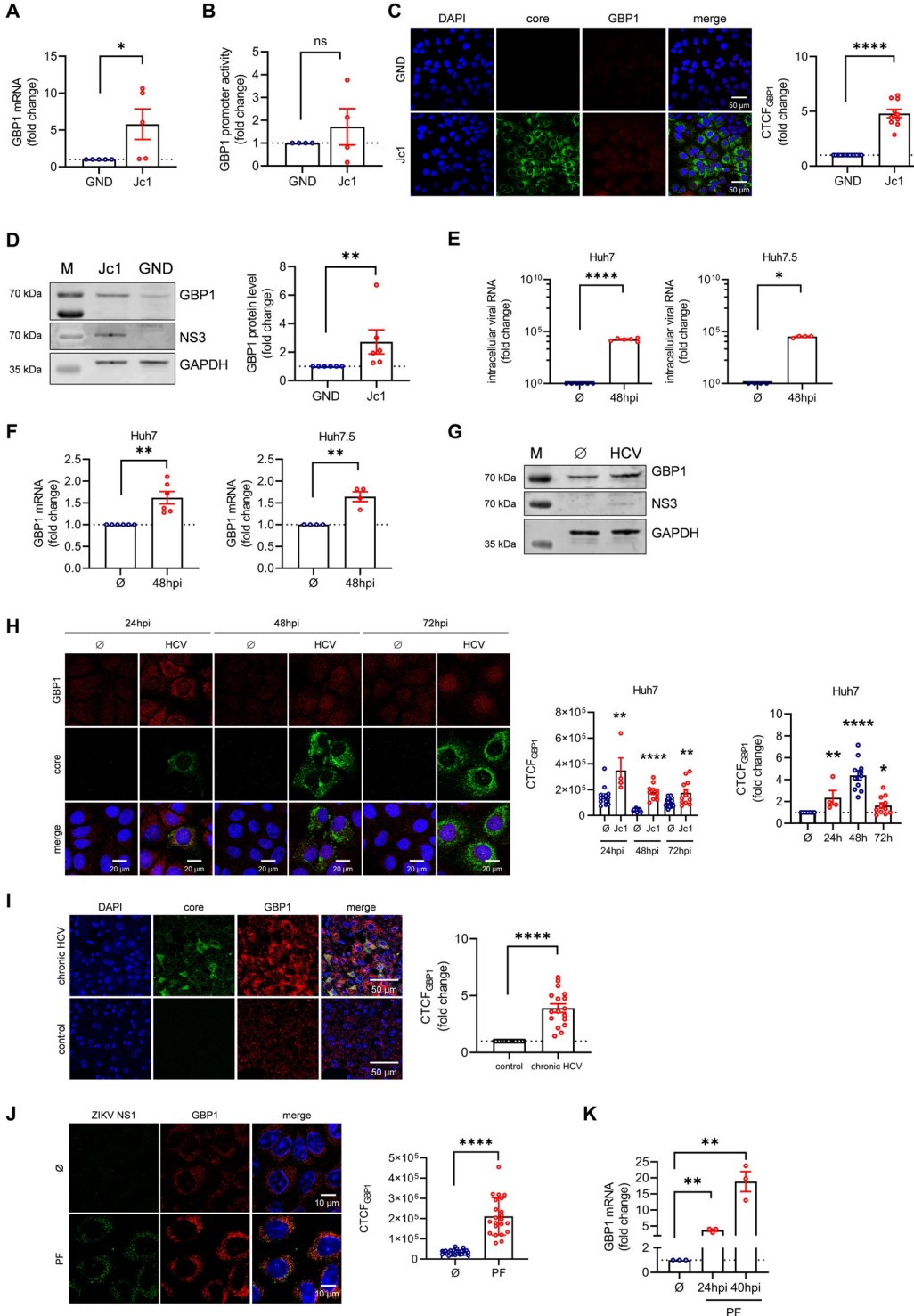

**Fig 1. The expression of GBP1 is increased in HCV replicating cells. (A)** qPCR analysis to monitor GBP1 levels in stably HCV replicating Huh7.5 cells (Jc1) and the corresponding negative cells (GND) 48 h post seeding. Relative change of the GBP1 mRNA levels were referred to the untreated (UT) GND cells (set to 1). N = 5 biological replicates. **(B)** Luciferase reporter gene assay of stably HCV replicating Huh7.5 cells (Jc1) and the corresponding negative cells (GND) transfected with pGL3-GBP1-Luc plasmid DNA (GBP1-Luc) for 48 h. Relative values are referred to the untreated control (GND (UT)) (set to 1). N = 4 biological replicates. **(C)** CLSM analysis of stably HCV replicating Huh7.5 cells (Jc1) and the corresponding HCV negative cells (GND) 48 h post seeding. GBP1 (red) and HCV core (green) were detected using

specific antisera. Nuclei were visualized with DAPI. Scale bar, 50 μm. Quantification of the relative CTCF (corrected total cell fluorescence) of GBP1. For each condition, a minimum of 10 cells was analyzed. Relative values are referred to the untreated control (GND) (set to 1). Images are representative of 3 independent experiments. **(D)** Representative Western blot and the respective densitometric analyses of stably HCV replicating Huh7.5 cells (Jc1) and the corresponding negative cells (GND). GBP1 and NS3, were detected using specific antisera. Detection of GAPDH served as loading control. For GBP1 relative protein expression was calculated. Graph shows the mean fold change of GBP1 protein levels relative to the negative cells (GND) (set to 1). N = 6 biological replicates. **(E-F)** qPCR analyses to monitor HCV **(E)** and GBP1 **(F)** RNA levels in uninfected (∅) and HCV infected Huh7 and Huh7.5 cells (GE = 100) after 48 h. Relative change of the GBP1 mRNA levels were referred to the uninfected (∅) cells (set to 1). Huh7, N = 6 technical replicates; Huh7.5, N = 4 technical replicates. **(G)** Western blot analysis of HCV infected Huh7 cells (GE = 100) and the corresponding uninfected cells (∅). GBP1 and NS3, were detected using specific antisera. Detection of GAPDH served as loading control. **(H)** CLSM analysis of uninfected (∅) or HCV infected Huh7 cells (GE = 100) after 24 h, 48 h, 72 h. GBP1 (red) and HCV core (green) were detected using specific antisera. Nuclei were visualized with DAPI. Scale bar, 20 μm. Quantification of the relative CTCF (corrected total cell fluorescence) of GBP1. For each condition, a minimum of 5 cells was analyzed. **(I)** CLSM analysis of liver sections derived from chronic HCV (genotype 1) infected patients and healthy livers as a control, using GBP1 (red) and HCV core (green) specific antisera. Nuclei were visualized with DAPI. Scale bar, 50 μm. Quantification of the relative CTCF (corrected total cell fluorescence) of GBP1. One representative sample is shown. For each condition, a minimum of 10 cells was analyzed. Relative values are referred to the healthy roughly sex- and age-matched control livers (set to 1). **(J)** CLSM analysis of uninfected (∅) or ZIKV infected A549 cells (French Polynesia (PF), MOI = 1) 12 h pi. GBP1 (red) and ZIKV NS1 (green) were detected using specific antisera. Nuclei were visualized with DAPI. Scale bar, 10 μm. Quantification of the relative CTCF (corrected total cell fluorescence) of GBP1. For each condition, a minimum of 10 cells was analyzed. Images are representative of 3 independent experiments. **(K)** qPCR analysis to monitor GBP1 levels in uninfected (∅) and ZIKV infected A549 cells (French Polynesia (PF), MOI = 1) after 24 h and 40 h. Relative change of the GBP1 mRNA levels were referred to the uninfected (∅) cells (set to 1). N = 3 biological replicates. **(A-K)** For all experiments, statistics were performed as mean ± SEM, unpaired t-test referred to ctrl. with *p<0.05, **p<0.01, ****p<0.0001, ns = not significant.

HCV-positive cells (Jc1) (**Fig 2F**). However, we could not observe any impact of the IFNγ-treatment on the actin structures in the Jc1 or GND cells (**Fig 2G**).

These data indicate that HCV attenuates the relative IFNγ-dependent induction on the protein level of GBP1 and the IFNγ-dependent GBP1 induction has no impact on the actin cytoskeleton.

## Silencing of GBP1 leads to intracellular accumulation of HCV core and to a reduction of NS3

To examine the crosstalk between HCV and GBP1 in more detail, the latter was silenced in HCV positive and the corresponding HCV negative cells using a scrambled control (scr RNA) or a GBP1-specific short interfering RNA (siRNA). The cells were harvested 72 h post siRNA transfection and the effect of decreased GBP1 levels was analyzed by quantitative CLSM-, qPCR- and Western blot analysis. As Huh7.5 cells express very low amounts of GBP1, the GBP1 knockdown can be hardly detected in the untreated (UT) cells. However, efficient silencing of GBP1 after stimulation with IFNγ was detected on protein and RNA levels (**Figs 3A, 3B** and **S2A**). To exclude a compensatory effect of GBP1 silencing by the expression of other GBPs, we analyzed the expression of GBP2/5, two GBP1-prenylation paralogs that are able to form heterodimers, by qPCR. No compensatory effect could be detected (**S2B and S2C Fig**). CLSM-based single-cell analyses via calculating the CTCF indicate that in GBP1-silenced cells(characterized by a low GBP1 level) the core signal was significantly increased (~50-fold) as compared to the scrambled control (scr RNA) (**Fig 3A**). In contrast to this, Western blot analyses indicate a slight decrease in the core protein (**Fig 3B**). This discrepancy could reflect the formation of poorly soluble core aggregates. Indeed, after GBP1 silencing, an intracellular accumulation of the core protein could be observed in the perinuclear region. This went along with an increased size of core-associated structures (**Fig 3C and 3D**). In contrast to the core-specific signal, silencing of GBP1 resulted in a decrease of the NS3 signal, as indicated by

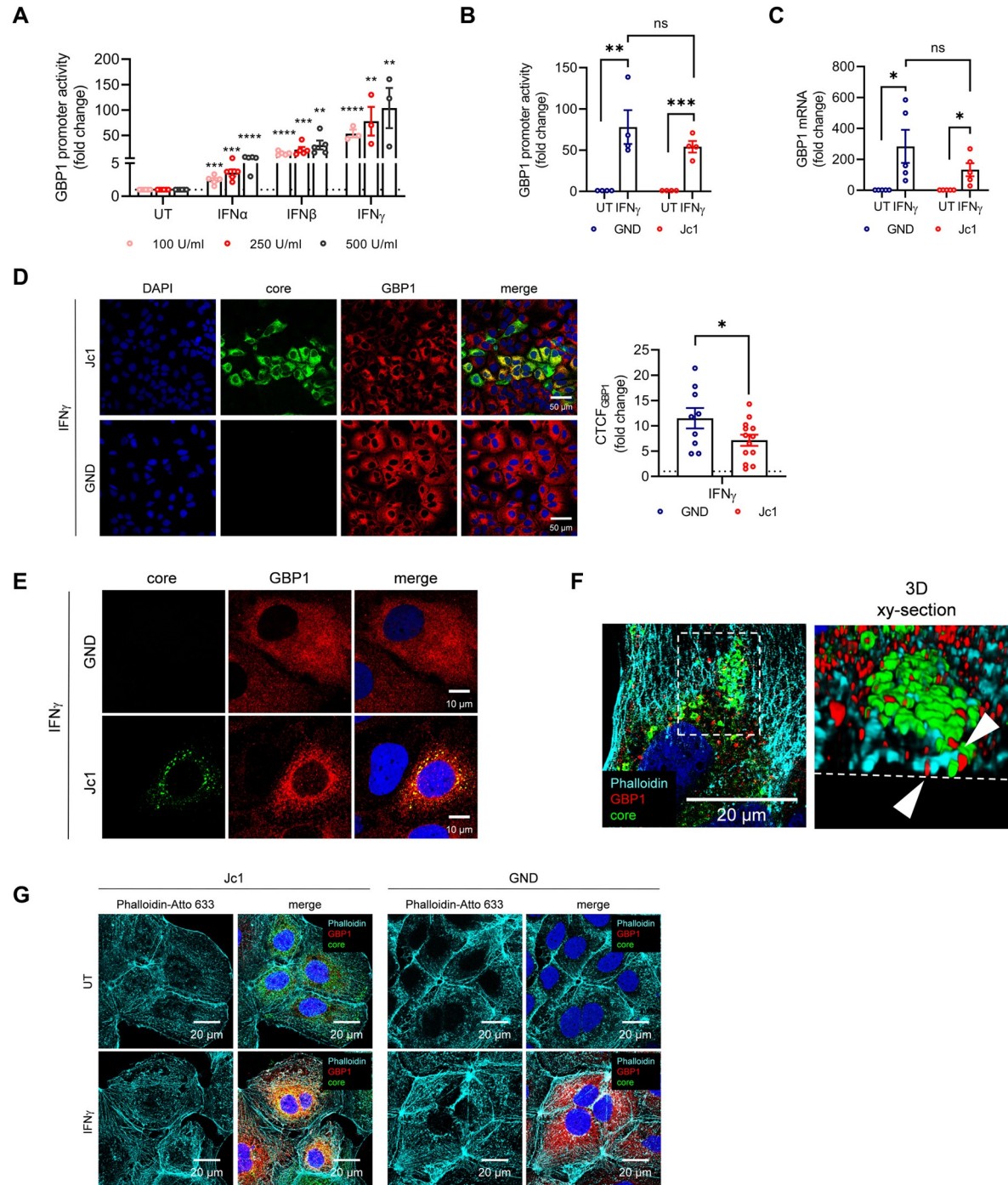

**Fig 2. The IFNγ-dependent induction of the GBP1 protein is impaired in HCV positive cells. (A)** Luciferase reporter gene assay of Huh7.5 cells transfected with pGL3-GBP1-Luc plasmid DNA (GBP1-Luc) to monitor GBP1 expression after treatment with different concentrations (100 U/ml, 250 U/ml, 500 U/ml) of IFNα, IFNβ, IFNγ for 24 h (48 h post seeding). Relative values are referred to the untreated control (UT) (set to 1). $N_{IFN\gamma}$ = 3 biological replicates; $N_{IFN\alpha, \, IFN\beta}$ = 5 biological replicates. **(B)** Relative induction of GBP1 promoter activity in stably HCV replicating (Jc1) and negative (GND) Huh7.5 cells after IFNγ-treatment (24h, 100 U/ml). Samples were analyzed 48 h post transfection (pt). Graph shows the mean fold change of the GBP1 promoter activity relative to the untreated (UT) controls (Jc1, GND) (set to 1). **(C)** Relative induction of GBP1 mRNA levels in stably HCV replicating (Jc1) and negative (GND) Huh7.5 cells after IFNγ-treatment (24 h, 100 U/ml). Samples were analyzed 48 h post seeding. Graph shows the mean fold change of GBP1 mRNA levels relative to the untreated (UT) controls (Jc1, GND) (set to 1). **(D)** CLSM analysis of stably HCV replicating Huh7.5 cells (Jc1) and the corresponding HCV negative cells (GND). 24 h post seeding the cells were treated with IFNγ (500 U/ml) for 24 h to induce GBP1 expression. GBP1 (red) and HCV core (green) were detected using specific antisera. Nuclei were visualized with DAPI. Scale bar, 50 μm.

Quantification of the relative CTCF (corrected total cell fluorescence) of GBP1 after IFNγ treatment. Relative values are referred to the untreated control (GND (UT)) (set to 1). For each condition, a minimum of 10 cells was analyzed. Images are representative of 3 independent experiments. (**E**) CLSM analysis of stably HCV replicating Huh7.5 cells (Jc1) and the corresponding HCV negative cells (GND). 24 h post seeding the cells were treated with IFNγ (100 U/ml) for 24 h to induce GBP1 expression. GBP1 (red) and HCV core (green) were detected using specific antisera. Nuclei were visualized with DAPI. Scale bar, 10 μm. (**F**) Z-stack and 3D-reconstruction (3D inset xy-section) of stably HCV replicating Huh7.5 cells (Jc1) to visualize actin structures that colocalize with GBP1 and HCV core, indicated by the white arrows. GBP1 (red), HCV core (green) and F-actin (cyan) were detected using specific antisera. Nuclei were visualized with DAPI. Scale bar, 20 μm. (**G**) CLSM analysis of stably HCV replicating Huh7.5 cells (Jc1) and the corresponding HCV negative cells (GND). 24 h post seeding the cells were treated with IFNγ (100 U/ml) for 24 h to induce GBP1 expression. GBP1 (red) and HCV core (green) and F-actin (cyan) were detected using specific antisera. Nuclei were visualized with DAPI. Scale bar, 20 μm. (**A-D**) For all experiments, statistics were performed as mean + SEM, unpaired t-test compared to ctrl. with *$p<0.05$, **$p<0.01$, ****$p<0.0001$.

CLSM-based single cell analysis (**Fig 3E**). Similarly, no significant effect on the distribution of the NS3 protein was observed (**Fig 3F**).

For a variety of viruses, GBP1 was described as a relevant factor driving the antiviral effect mediated by IFNγ. In contrast to this, CLSM analyses on a single-cell level via calculating the CTCF revealed that the IFNγ-dependent reduction of the NS3-specific signal was even more pronounced in the siGBP1-treated cells as compared to the scr-control (**Fig 3E**). In line with this, silencing of GBP1 did not affect the IFNγ-dependent reduction of intracellular HCV genomes (**Fig 3G**).

In addition, we analyzed the impact of GBP1 silencing on the actin cytoskeleton. Again, we could not detect any impact of GBP1 modulation on the actin filaments (**Fig 3H**).

Taken together, these data indicate that silencing of GBP1 leads to a decreased amount of NS3 and leads to an intracellular accumulation of HCV core structures with an increased size. Moreover, the antiviral effect of IFNγ against HCV does not depend on GBP1.

## Silencing of GBP1 affects lysosomal functionality and leads to a reduction in specific infectivity of HCV viral particles

The observed accumulation of core suggests that silencing of GBP1 has an impact on HCV morphogenesis. As HCV morphogenesis and release are tightly associated with autolysosomal structures [47,11], we analyzed the impact of GBP1 silencing on LAMP2 positive structures. In GBP1 positive cells, we found LAMP2 in close proximity to NS3 in the perinuclear region, together with GBP1 (**Fig 4A**). Interestingly, we observed a changed staining pattern of LAMP2 vesicles upon silencing of GBP1, which went along with a reduced LAMP2 amount after IFNγ stimulation (**Fig 4C**). With respect to the HCV core protein, 3D reconstructions illustrate that the core protein localizes around these structures (**Fig 4B**), whereas NS3 can be found within LAMP2 positive structures (**Fig 4B**). To study whether GBP1-silencing has an impact on the lysosomal functionality and acidification, CLSM analyses of cells stained with LysoTracker, HCV core, GBP1 and LAMP2 was performed. These single-cell analyses revealed that GBP1-silencing impairs lysosomal acidification, as reflected by a decrease of the LysoTracker signal and an increase in the LAMP2-specific signal. This results in a decreased LysoTracker-LAMP2 ratio (**Fig 4C**).

LAMP2 is an essential factor involved in maturation of autophagosomes. In line with this, HCV hijacks the autophagosomal pathway for efficient release of viral particles [11]. To further determine the impact of GBP1 silencing on viral titers in this context, we determined the TCID$_{50}$ upon modulation. In contrast to previous data that suggest an antiviral effect of GBP1, our data revealed that silencing of GBP1 is associated with autophagosomal/lysosmal dysfunction and resulted in a significant reduction of intra- and extracellular viral titers. Ultimately, this indicates a pro-viral role of GBP1 for the release of infectious viral particles (**Fig 4D**). However, no significant effect on the number extracellular viral genomes was observed

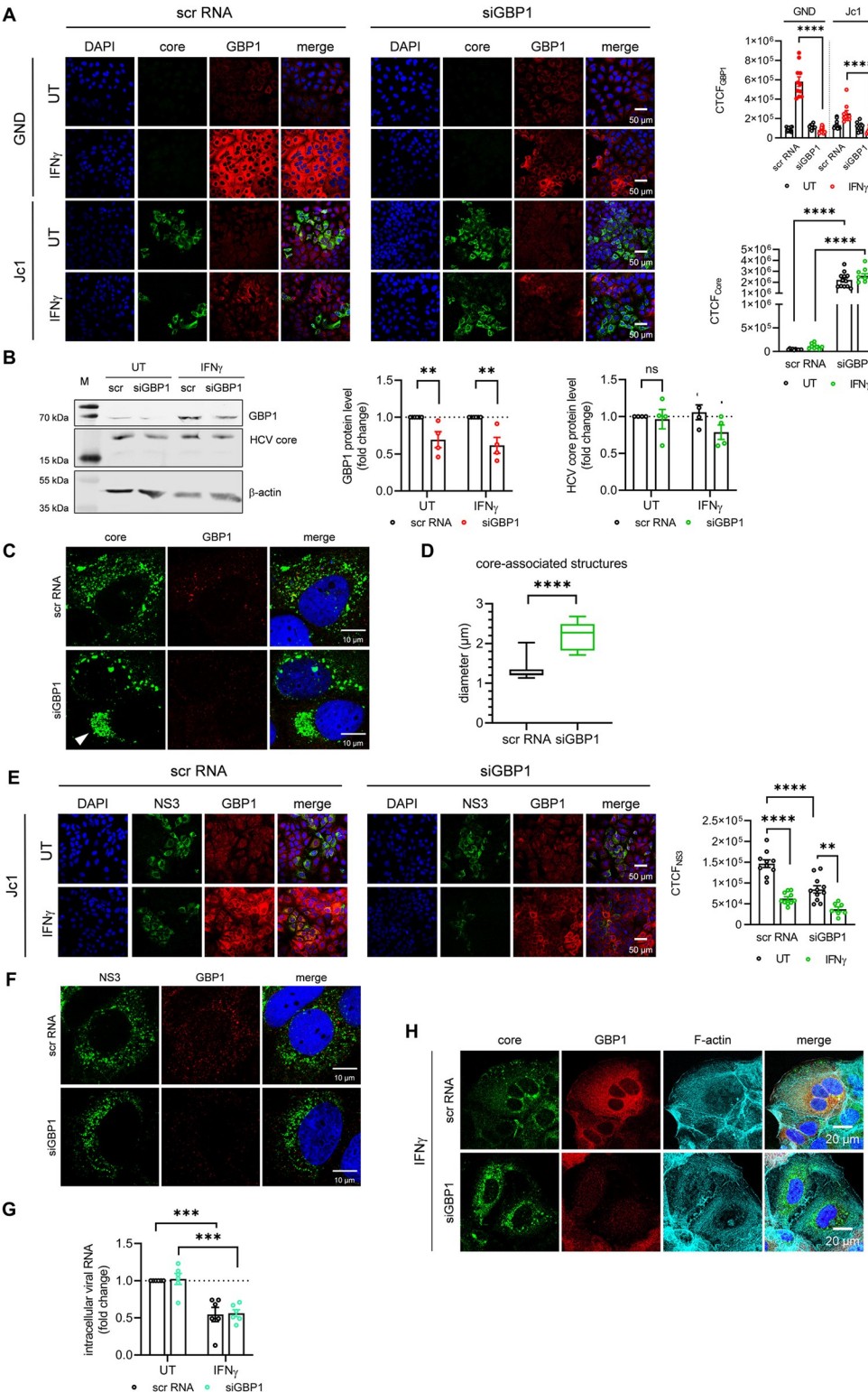

**Fig 3. Silencing of GBP1 leads to intracellular accumulation of HCV core and to a reduction in HCV NS3. (A)** CLSM analysis of stably HCV replicating Huh7.5 cells (Jc1) and the corresponding HCV negative cells (GND) 72 h after GBP1 silencing. GBP1 (red) and HCV core (green) were detected using specific antisera. Nuclei were visualized with DAPI. Scale bar, 50 μm. Quantification of the relative CTCF (corrected total cell fluorescence) of GBP1 and HCV core. For each condition, a minimum of 10 cells was analyzed. Images are representative of 4 biological replicates. **(B)**

Representative Western blot and the respective densitometric analyses of stably HCV replicating Huh7.5 cells (Jc1) 72 h after GBP1 silencing. GBP1 and HCV core, were detected using specific antisera. Detection of beta-actin served as loading control. For GBP1 relative protein expression was calculated. Graph shows the mean fold change of GBP1 protein levels relative to the scr RNA controls (UT, IFNγ) (set to 1). Relative HCV core values are referred to the (UT) scrambled (scr) control (set to 1). N = 4 biological replicates **(C)** CLSM analysis of stably HCV replicating Huh7.5 cells (Jc1) 72 h after GBP1 silencing. GBP1 (red) and HCV core were detected using specific antisera. Nuclei were visualized with DAPI. Scale bar, 10 μm. Images are representative of 4 biological replicates. **(D)** Calculation of the average diameter of HCV core-associated structures after GBP1 silencing. Box plots, box and whiskers show min to max, line shows the mean value (N = 9 cells/treatment). Statistics: Mean + SEM, unpaired t-test compared to scr RNA with *p<0.05, **p<0.01, ****p<0.0001. **(E, F)** CLSM analysis of stably HCV replicating Huh7.5 cells (Jc1) and the corresponding HCV negative cells (GND) 72 h after GBP1 silencing. GBP1 (red) and NS3 (green) were detected using specific antisera. Nuclei were visualized with DAPI. Scale bar, 50 μm (E), 10 μm. Images are representative of 4 biological replicates. Quantification of the relative CTCF (corrected total cell fluorescence) of NS3 (E). For each condition, a minimum of 10 cells was analyzed. **(G)** qPCR analyses to detect intracellular HCV genomes 72 h after GBP1 silencing. Relative change of the HCV genomes were referred to the untreated (UT) scrambled (scr) control (set to 1). N = 6 biological replicates. **(H)** CLSM analysis of stably HCV replicating Huh7.5 cells (Jc1) after GBP1 silencing. GBP1 (red) and HCV core (green) and F-actin (cyan) were detected using specific antisera. Nuclei were visualized with DAPI. Scale bar, 20 μm. **(A, B, E, G)** To induce GBP1 expression the cells were treated with IFNγ (100 U/ml) for 24 h or left untreated as control (UT). For experiment **(B)**, statistics were performed as mean ± SEM, unpaired t-test compared to ctrl., for experiment **(A, E, G)** were performed as mean ± SEM, one-way ANOVA with *p<0.05, **p<0.01, ****p<0.0001, ns = not significant.

(**Fig 4E**). As a consequence, the specific infectivity, described by the ratio of infectious viral particles (TCID$_{50}$/ml) to viral genomes (genomes/ml), is decreased upon silencing of GBP1 (**Fig 4F**). Again, the antiviral effect of IFNγ on HCV—here on the titers of infectious HCV particles–is not abrogated by the loss of GBP1 (**Fig 4D**).

To further investigate whether the observed effect of GBP1 silencing on the lysosomal acidification can be rescued by induction of autophagy, we treated the cells Rapamycin for 16 h. Rapamycin inhibits the mammalian target of rapamycin complex 1 (mTORC1), thus activating the ULK1/2 complex, which induces autophagy. CLSM-based single cell analysis revealed that Rapamycin treatment restores the LAMP2- and LysoTracker-specific signal in the siGBP1 transfected cells. Consequently, Rapamycin treatment increased the LysoTracker-LAMP2 ratio after GBP1 silencing (**Fig 4G**). Moreover, induction of autophagy increased the HCV core-specific signal (**Fig 4G**).

Overall, these data indicate that GBP1 is a relevant factor for morphogenesis, trafficking or release of the viral particles.

## Lysosomal targeting of HCV core does not depend on GBP1

For HEV, it has been described that GBP1 targets HEV capsids towards lysosomal degradation [25]. Indeed, the HCV life cycle is closely connected to the autophagosomal pathway [48–50,11,51]. The latter represents a branching point between targeting virions either for release or for lysosomal degradation. In light of this, we speculated whether GBP1 modulates the targeting of HCV towards release or lysosomal degradation. To address this question, we performed CLSM analyses of HCV-positive cells either transfected with a GBP1-Wt encoding plasmid or treated with IFNγ to induce GBP1 expression. These were then stained for GBP1, HCV core and the lysosomal marker LAMP2 (**Fig 5A**). IFNγ-treatment strongly increased the GBP1- and LAMP2 signal and resulted in enlarged lysosomal structures. However, only a small fraction of HCV core could be detected within these LAMP2 structures as indicated by the profile plots (**Fig 5B**). GBP1-Wt transfection, independent from IFNγ, had no effect on the LAMP2 distribution and rather decreased the LAMP2-specific signal. Notably, induction of GBP1 expression, either by transfection with a GBP1-Wt construct or by IFNγ-treatment, resulted in high amounts of GBP1 colocalizing with HCV core in the perinuclear region (**Fig 5A**). To block lysosomal degradation, we treated HCV-positive cells transfected with the

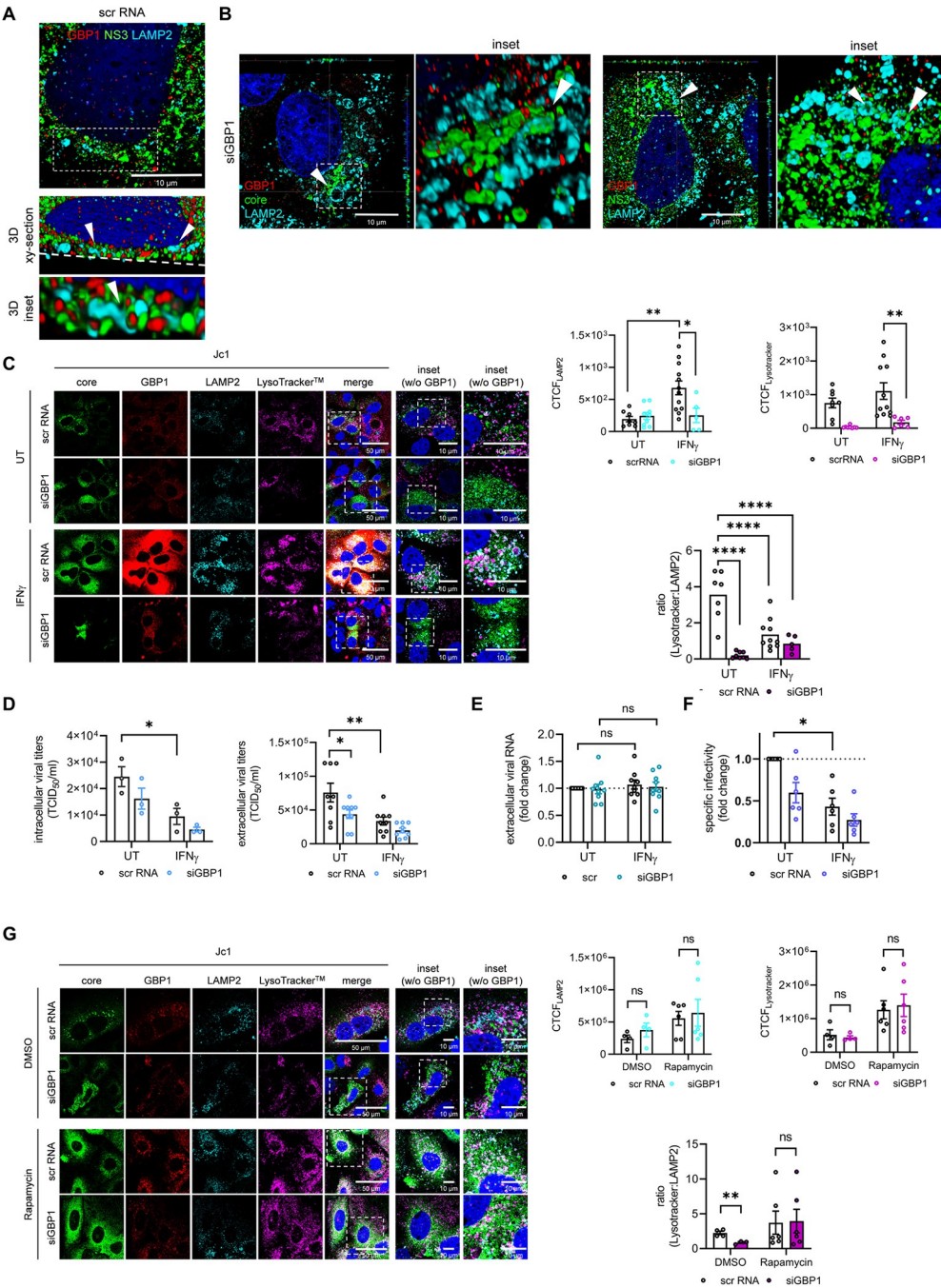

**Fig 4. Silencing of GBP1 affects lysosomal functionality and leads to a reduction in specific infectivity of HCV viral particles. (A)** Z-stack and 3D-reconstruction (3D inset xy-section, 3D inset) of stably HCV replicating Huh7.5 cells (scr RNA) to visualize LAMP2 structures colocalizing with NS3 and GBP1 as part of the RCs, indicated by the white arrows. GBP1 (red), NS3 (green) and LAMP2 (cyan) were detected using specific antisera. Nuclei were visualized with DAPI. Scale bar, 10 μm. Images are representative of 3 biological replicates. **(B)** Z-stack and 3D-reconstruction (inset) of HCV-replicating Huh7.5 cells 72 h after GBP1 silencing to visualize HCV core stacking outside and NS3 inside lysosomal structures, as indicated by the white arrows. GBP1 (red), HCV core/ NS3 (green) and LAMP2 (cyan) were detected using specific antisera. Nuclei were visualized with DAPI. Scale bar, 10 μm. Images are representative of 3 biological replicates. **(C)** CLSM analysis of stably HCV replicating Huh7.5 cells (Jc1) 72 h after GBP1 silencing. LysoTracker Deep Red staining (magenta) was performed for 1h at 37°C before fixing the cells with 4% FA. GBP1 (red), HCV core (green) and LAMP2 (cyan) were detected using specific antisera. Nuclei were visualized with DAPI. Scale bar, 50 μm (inset 10 μm). Quantification of the relative CTCF (corrected total cell fluorescence) of LAMP2 and LysoTracker. For each condition, a minimum of 10 cells was analyzed. To detect the impact of GBP1 silencing on

lysosomal acidification the ratio LAMP2/Lysotracker was calculated. Images are representative of 3 independent experiments. All immunofluorescence images were deconvoluted using the LasX Lightning Tool using the adaptive algorithm. **(D)** Detection of intra- and extracellular viral titers by $TCID_{50}$ 72 h after GBP1 silencing. N = 3 (intracellular), 9 (extracellular) biological replicates. **(E)** Detection of extracellular viral genomes by qPCR 72 h after GBP1 silencing. Relative change of the extracellular HCV genomes were referred to the untreated (UT) scrambled (scr) control (set to 1). N = 6 biological replicates. **(F)** The specific infectivity after GBP1 silencing was determined by calculating the ratio of the number of extracellular infectious viral particles ($TCID_{50}$/ml), measured by $TCID_{50}$, and the total amount of extracellular viral genomes (genomes/ml) determined by qPCR. **(G)** CLSM analysis of stably HCV replicating Huh7.5 cells (Jc1) 72 h after GBP1 silencing. To induce autophay the cells were treated with Rapamycin (100 nM) 16 h before harvesting. LysoTracker Deep Red staining (magenta) was performed for 1h at 37°C before fixing the cells with 4% FA. GBP1 (red), HCV core (green) and LAMP2 (cyan) were detected using specific antisera. Nuclei were visualized with DAPI. Scale bar, 50 μm (inset 10 μm). Quantification of the relative CTCF (corrected total cell fluorescence) of LAMP2 and LysoTracker. For each condition, a minimum of 4 cells was analyzed. To detect the impact of GBP1 silencing on lysosomal acidification the ratio LAMP2/Lysotracker was calculated. All inset immunofluorescence images were deconvoluted using the LasX Lightning Tool using the adaptive algorithm. For all experiments, statistics were performed as mean ± SEM, one-way ANOVA with *p<0.05, **p<0.01, ****p<0.0001, ns = not significant.

wildtype GBP1 with leupeptin in order to trap and visualize HCV core within lysosomal structures (**Fig 5C**). Successful Leupeptin treatment is indicated by an increased p62 signal (**S3 Fig**). Colocalization of HCV core with lysosomal structures was analyzed by thresholded Mender's overlap coefficient (tMOC) (**Fig 5D**). Single-cell analyses revealed a fraction of HCV core residing within lysosomal structures, as reflected by the profile (**Fig 5E**). Notably, the immunofluorescence data indicate that GBP1 colocalized with HCV core- and LAMP2-positve structures. However, core was also found in lysosomal structures independent from GBP1 expression (**Fig 5F**).

Based on these observations, we assume that GBP1 is not relevant for targeting of HCV particles towards the lysosomal compartment. This is in accordance to the observation, that (i) GBP1 has no detrimental effect on the release of HCV and (ii) GBP1 is not required for the antiviral effect of IFNγ.

## GBP1 overexpression promotes formation and release of infectious viral particles

The significantly decreased number of extracellular viral particles found in the context of GBP1 silencing suggest a potential pro-viral activity of GBP1. To further evaluate the role of GBP1 for the HCV life cycle, we analyzed the impact of GBP1 overexpression on HCV-positive cells. Therefore, the cells were transfected with a plasmid encoding GBP1-Wt or a control plasmid (pCMV-2b, Mock). Western blot analysis revealed a strong increase in GBP1 levels and a significant increase in the amount of NS3 as compared to the control. Oppositely, the amount of HCV core was significantly reduced (**Fig 6A**). In accordance to the decreased amount of HCV core, a significantly decreased number of intracellular genomes and a slight reduction of intracellular infectious viral particles was observed. The number of extracellular genomes and released infectious particles, however, was significantly increased (**Fig 6B–6E**). This might reflect that GBP1, when overexpressed, favors release of viral particles, which is associated with an intracellular decrease of genomes.

To further determine the impact of GBP1 overexpression on HCV proteins, we performed CLSM analyses (**Fig 6F**) using GBP1-, and core-specific antibodies. In Mock-transfected cells, the core protein was equally distributed over the cell, while an enrichment in larger perinuclear regions, partially colocalizing with GBP1 (as illustrated in the inset), were observed (**Fig 6F**). Overexpression of GBP1 displays a vesicular-like distribution in the perinuclear region. Here, the core protein showed a strong colocalization with GPB1 around the nucleus (**Fig 6F**).

Taken together, the data presented above reflect that GBP1 plays a role in the later steps of the viral life cycle and favors viral release.

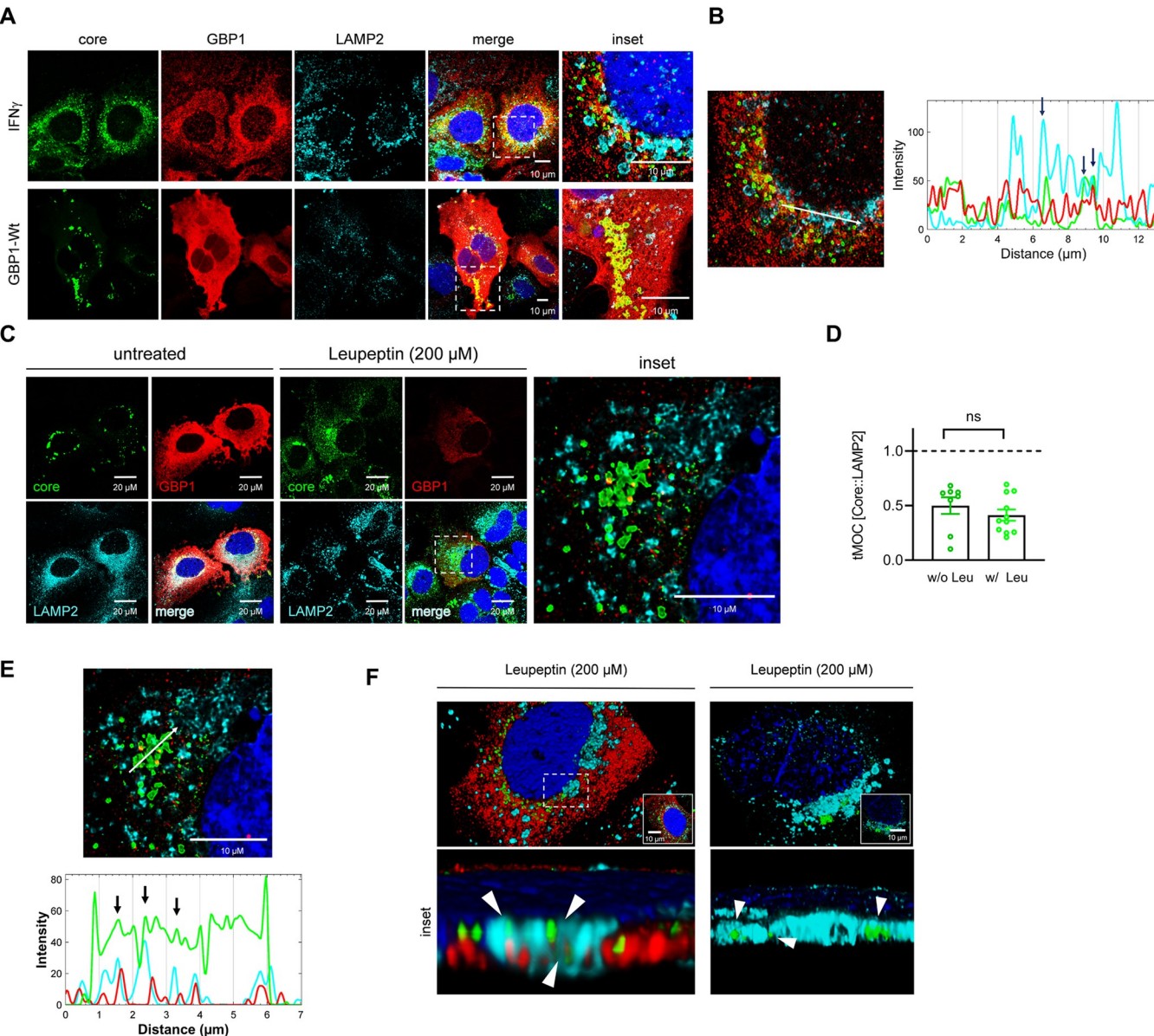

**Fig 5. Lysosomal targeting of HCV core does not depend on GBP1. (A)** CLSM analysis of stably HCV replicating Huh7.5 cells using GBP1-, HCV core-, and LAMP2-specific antisera. To induce GBP1 expression the cells were treated with IFNγ (100 U/ml) for 24 h (upper panel) (24 post seeding) or transfected with a GBP1-Wt encoding construct for 48 h (lower panel). Scale bars, 10 μm. Images are representative of at least 5 independent experiments. **(B)** Profile Plot corroborating that only a minor fraction of HCV core is found within lysosomal structures in cells with high GBP1 amounts, as indicated by the black arrows. **(C)** CLSM analysis of stably HCV replicating Huh7.5 cells transfected with a GBP1-Wt encoding construct using GBP1-, HCV core- and LAMP2-specific antisera. To inhibit lysosomal degradation the cells were treated with Leupeptin (200μM) for 24 h or left untreated as a control (UT). The cells were analyzed 48 h pt. Scale bars, 20 μm (inset 10 μm). Images are representative of 3 independent experiments. **(D)** tMOC as calculated for the HCV core (green) signal overlapping with the LAMP2 (cyan) signal with (w/) or without (w/o) Leupeptin (Leu) treatment. **(E)** Profile Plot corroborating that Leu-treatment does not increase HCV core amounts within lysosomal structures (indicated by the black arrows). **(F)** 3D-reconstruction of GBP1-Wt transfected stably HCV positive cells (Jc1) that were treated with Leupeptin (Leu) (200μM). The white arrows indicate HCV core within LAMP2-positive structures (cyan) independent of GBP1 expression. Scale bar, 10 μm. All immunofluorescence images were deconvoluted using the LasX Lightning Tool using the adaptive algorithm.

## GBP1 colocalizes with HCV core on the surface of lipid droplets

The above described data show a strong pro-viral impact of GBP1 on the HCV life cycle. To further characterize this, the crosstalk between GBP1 and HCV, with respect to the localization of GBP1 in HCV-positive cells, was analyzed in detail. Data set forth above unveiled an altered

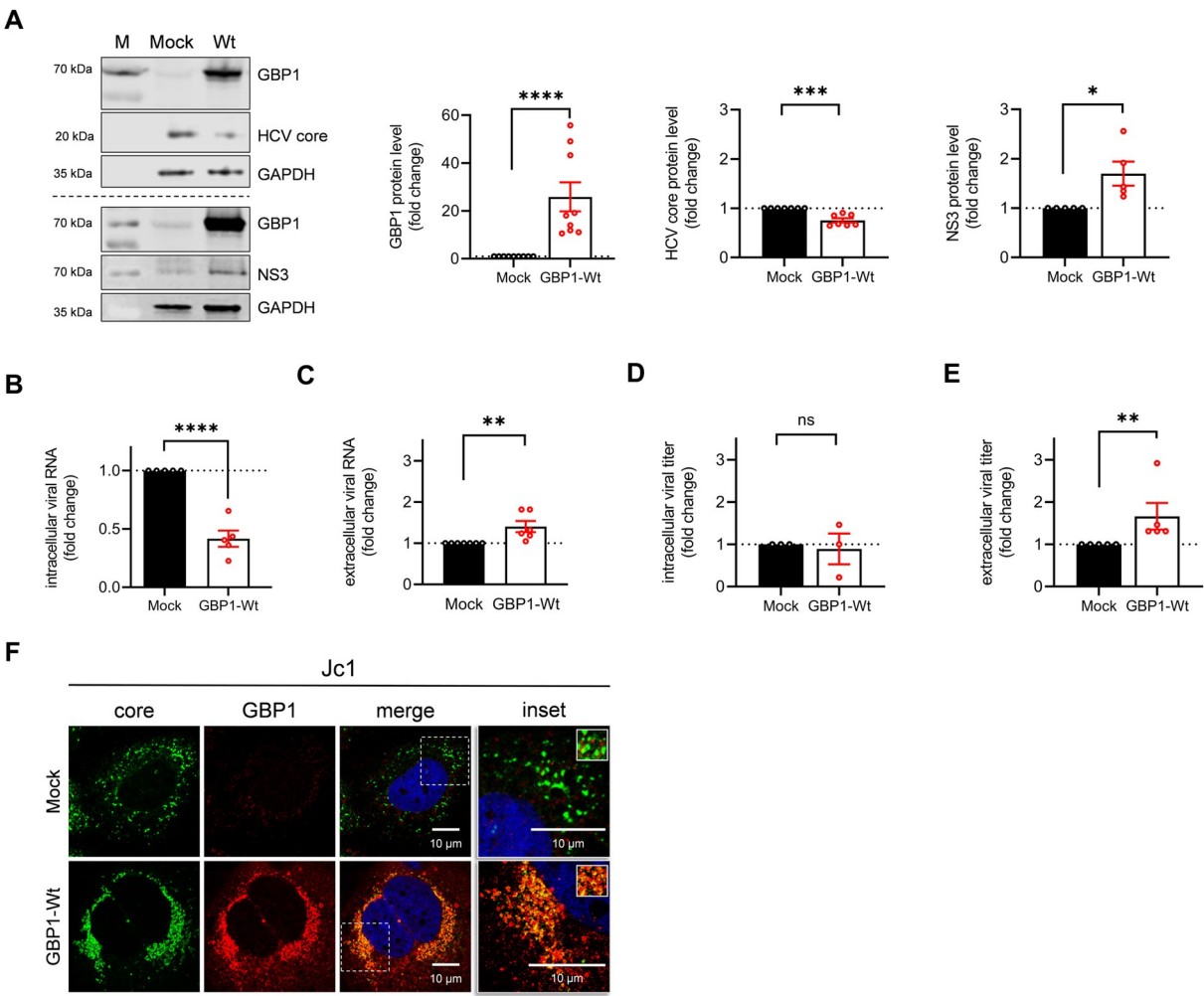

**Fig 6. GBP1 overexpression promotes formation and release of infectious viral particles.** Stably HCV replicating Huh7.5 cells (Jc1) were transfected with the GBP1-Wt construct or the empty vector (pCMV-2B) (Mock) for 48 h. **(A)** Representative Western blot and the respective densitometric analysis. GBP1, HCV core, and NS3 were detected using specific antisera. Detection of GAPDH served as loading control. Relative values are referred to the Mock control (set to 1). $N_{Core} = 8$, $N_{GBP1} = 9$, $N_{NS3} = 5$ biological replicates. For all experiments, statistics were performed as mean ± SEM, for GBP1 and HCV core using Mann-Whitney test referred to ctrl. with **$p < 0.01$, ****$p < 0.0001$ and NS3 using unpaired t-test compared to ctrl. with *$p < 0.05$. **(B)** qPCR analysis to monitor intracellular HCV genomes. Relative values are referred to the empty vector (Mock) (set to 1). N = 4 biological replicates. Statistics were performed as mean ± SEM, unpaired t-test compared to ctrl. with ****$p < 0.0001$. **(C)** Detection of extracellular viral genomes by qPCR. Relative values are referred to the empty vector (Mock) (set to 1). N = 6 biological replicates. Statistics were performed as mean ± SEM, unpaired t-test compared to ctrl. with **$p < 0.01$. **(D, E)** Quantification of intra- and extracellular viral titers. Relative values are referred to the empty vector (Mock) (set to 1). N = 4 biological replicates. Statistics were performed as mean ± SEM, Mann-Whitney test compared to ctrl. with **$p < 0.01$. **(F)** CLSM analysis. GBP1 (red) and HCV core (green) were detected using specific antisera. Nuclei were visualized with DAPI. To better visualize the GBP1 distribution the signal of the red channel was decreased. Scale bar, 10 μm. The additional zoom factor was 3.5. Images are representative of at least 3 independent experiments. All immunofluorescence images were deconvoluted using the LasX Lightning Tool using the adaptive algorithm.

localization of GBP1 in HCV replicating cells. Infection with HCV resulted in a perinuclear accumulation of GBP1, which could not be observed in the GND control cells, where GBP1 is equally distributed in the cytoplasm. 3D reconstructions of Jc1 cells overexpressing GBP1-Wt indicate a vesicular distribution of GBP1. These structures were found in close proximity to HCV core proteins (tMOC 60%) and LAMP2 positive vesicles in the perinuclear region (**Fig 7A**).

Due to the impact of GBP1 on HCV morphogenesis and release and the relevance of lipid droplets (LDs) for HCV morphogenesis, we analyzed whether the above described

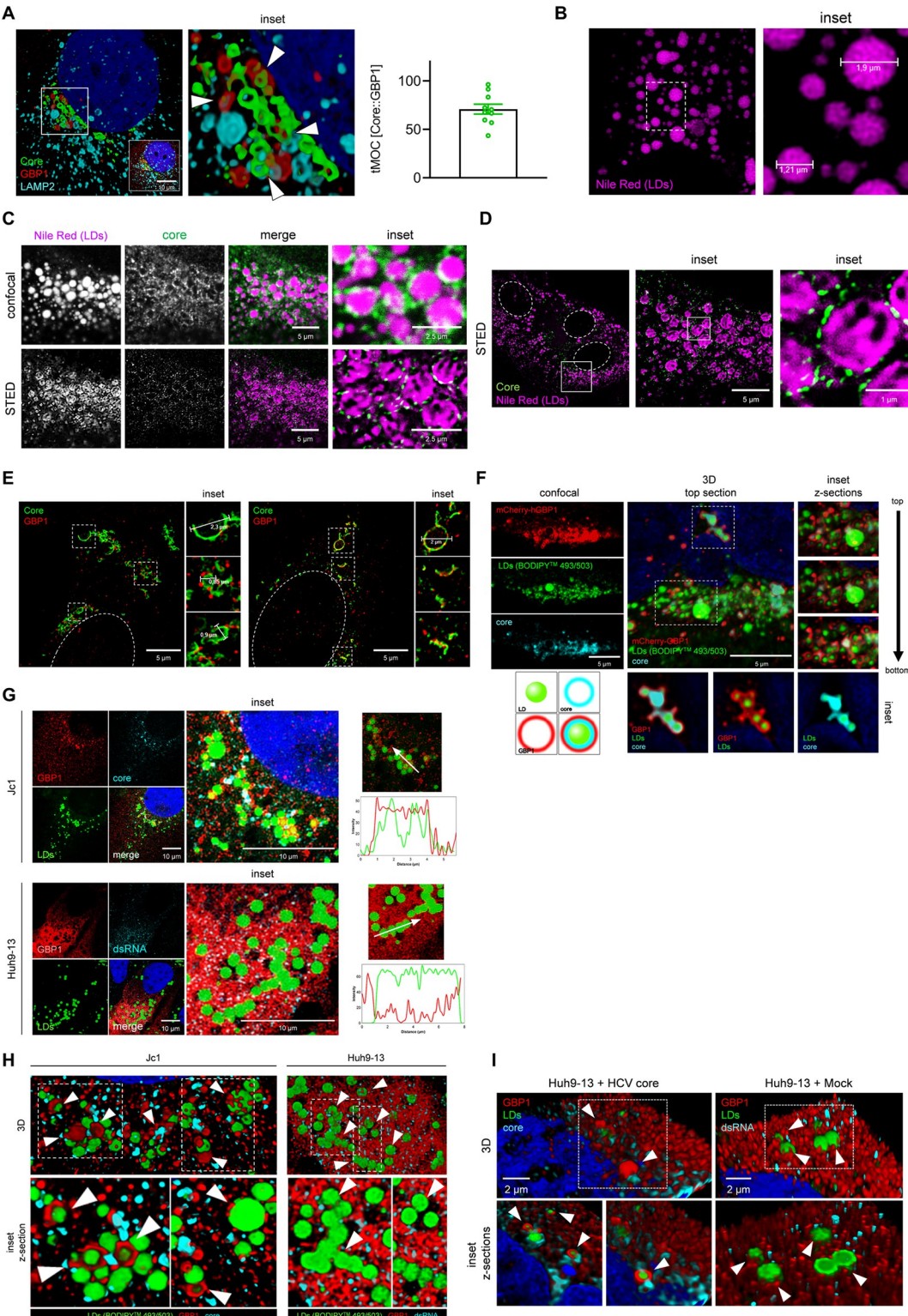

**Fig 7. GBP1 colocalizes with HCV core on the surface of lipid droplets. (A)** 3D-reconstruction of GBP1-Wt transfected stable HCV positive Huh7.5 cells (Jc1) (48 h pt) stained for GBP1 (red), HCV core (green) and LAMP2 (cyan). The white arrows indicate HCV core, GBP1 and LAMP2 vesicles in close proximity. Scale bar, 10 μm. tMOC was calculated for the HCV core (green) signal overlapping with the GBP1 (red) signal. **(B)** CLSM analysis of stable HCV positive Huh7.5 cells (48 h post seeding) to visualize LDs using Nile Red (magenta). The size of the LDs varies approximately between 1–2μm. Images are

representative of 3 independent experiments. **(C, D)** CLSM (upper panel) (C) and dual color STED images (lower panel) (C, D) of stable HCV positive Huh7.5 cells to visualize HCV core on the surface of LDs. HCV core (green) was detected using specific antisera labeled with Atto594. LDs were visualized using Nile Red (magenta). Scale bar, 5 μm (inset 2.5 μm (C), 1 μm (D)). Images are representative of 3 independent experiments. **(E)** Dual color STED images of stable HCV positive cells. To induce GBP1 expression the cells were treated with IFNγ (500 U/ml, (left); 100 U/ml (right)) for 24h. The cells were analyzed 48 h post seeding. GBP1 (red) and HCV core (green) were detected using specific antisera. Scale bar, 5 μm. Images are representative of at least 3 independent experiments. **(F)** CLSM analysis of stable HCV-positive Huh7.5 cells transfected with a mCherry-GBP1 (red) construct for 48 h) stained for LDs (BODIPY 495/503) (green) and HCV core (cyan). 3D reconstruction (middle) and different sections (from top to bottom) of the z-stack to visualize GBP1 surrounding LDs (right). Scale bar, 5 μm. Graphical illustration to visualize the vesicular-like distribution of GBP1 around HCV core coated LDs. **(G)** CLSM analysis of stable HCV positive Huh7.5 cells (Jc1) stained for GBP1 (red), HCV core (cyan), and LDs (BODIPY 495/503) (green) (left) and the subgenomic cell line Huh9-13 stained for GBP1 (red), dsRNA (cyan) and LDs (BODIPY 495/503) (green) (right). To induce GBP1 expression the cells were treated with IFNγ (100 U/ml) for 24 h. The cells were analyzed 48 h post seeding. Scale bar, 10 μm. **(H)** 3D reconstructions and profile plots corroborate the localization of GBP1 on the surface of LDs in the HCV positive cells that could not be detected in the subgenomic Huh9-13 cells. All immunofluorescence images were deconvoluted using the LasX Lightning Tool using the adaptive algorithm. **(I)** 3D reconstructions and z-sections of the subgenomic cell line Huh9-13 transfected with an HCV core encoding plasmid (pcCore) or the corresponding control (Mock, pcDNA3.1.(-)) for 48 h stained for GBP1 (red), dsRNA/HCV core (cyan) and LDs (BODIPY 495/503) (green). To induce GBP1 expression the cells were treated with IFNγ (100 U/ml) for 24 h. Scale bar, 2 μm. All immunofluorescence images were deconvoluted using the LasX Lightning Tool using the adaptive algorithm.

GBP1-vesicles represent structures of the membranous web involved in viral morphogenesis. First, we performed CLSM analyses of LDs exclusively to define the size of LDs (**Fig 7B**). To precisely define the HCV core-GBP1-associated structures, we further performed super resolution microscopy using dual color STED imaging and either co-stained for HCV core and LDs (**Fig 7C and 7D**), or HCV core and GBP1 (**Fig 7E**). As expected, the HCV core protein could be detected around LDs (**Fig 7C and 7D**). In addition, we found GBP1 colocalizing with the core protein in vesicular-shaped structures that displayed the same size as the HCV core-deco-rated LDs (1–2 μm) (**Fig 7B and 7E**). In line with this, CLSM analysis, including z-stacking and 3D reconstructions of HCV positive cells transfected with a mCherry-hGBP1 construct, clearly illustrated that the HCV core protein covers the surface of the LDs. This is consistent with previous studies [52]. More importantly, we found mCherry-hGBP1 in a concentric staining pattern on the surface of the LDs and in association with the HCV core protein (**Fig 7F**). GBP1 has been described to associate with intracellular membranes and organelles [18,21]. To address whether targeting of GBP1 towards LDs depends on the presence of the HCV core protein, we performed CLSM analyses including z-stacking and 3D reconstructions of HCV positive cells and the subgenomic cell line Huh9-13. The latter stably replicate HCV, but lack the structural proteins and therefore do not produce infectious particles. Huh9-13 and Huh7.5 cells produce GBP1 in a comparable level. To induce GBP1 expression, the cells were treated with IFNγ and co-stained for HCV core (Jc1) or dsRNA (Huh9-13), GPB1 and LDs. The profile plot and 3D reconstructions of HCV positive cells demonstrated that GBP1 is found on the surface of the LDs. Interestingly, we observed areas where it completely coats the surface of the LDs. However, in the Huh9-13 cells lacking the HCV core protein, GBP1 was found equally distributed throughout the whole cytoplasm in a reticular staining pattern. Here, only partial spiking of the LDs was observed, which could be mediated by the C-terminal farnesyl residue that mediates association with lipids (**Fig 7G and 7H**). Importantly, transfec-tion of Huh9-13 cells with a plasmid encoding HCV core resulted in the same GBP1 pheno-type as in the Jc1 cells, with GBP1 completely coating the LDs (**Fig 7I**). The observations above imply a direct HCV core-GBP1 interaction. To test this, we performed coimmunoprecipita-tion experiments (co-IP) using a Flag-tagged GBP1 construct. However, we were not able to detect a direct association of these proteins under these conditions, suggesting that additional factors at the membrane environment are required to mediate the association of GBP1 and HCV core. Although the proof of a direct interaction by co-IP failed, the super resolution data

imply a functional interaction of HCV core and GBP1 on the surface of LDs, which might serve as an assembly platform.

These data indicate that GBP1 localizes to surface of LDs in HCV-positive cells. Here, it forms a concentric staining pattern together with the HCV core protein. Interestingly, despite no direct GBP1-core association could be detected, the targeting of GBP1 to the LDs depends on the presence of HCV core.

## Discussion

Infection with HCV triggers the production of type I and II IFNs, which display antiviral and immune regulatory activities mediated through expression and induction of ISGs. There is a variety of reports describing an antiviral effect of the IFN-inducible GTPases GBPs on multiple viruses [22].

For HCV, a decreased expression of GBP1 was described in cells harboring a subgenomic HCV replicon construct [27]. Further analyses of the same group identified an interaction of the viral polymerase NS5B finger domain with the N-terminal GTPase domain of GBP1, which blocks its antiviral activity [28].

In contrast to these data, we observe a proviral effect of GBP1 on HCV. Unlike recently reported, we found an increased GBP1 expression (~7-fold) and protein amount in HCV replicating cells. Moreover, the IFNγ-dependent induction of GBP1 is slightly attenuated in HCV-positive cells. As HCV has evolved strategies to counteract IFN-dependent signaling processes, we assume that this observation reflects the interference of HCV proteins with factors involved in JAK-STAT signaling pathways or by inhibition of cRaf-dependent signaling cascades [37,12,53]. Certainly, our data indicate a yet unidentified new role of GBP1 during HCV infection. However, pro-viral effects of GBP1 are not unprecedented.

Recently, a pro-viral role of GBP1 during JEV infection has been described. This is based on inhibition of the innate immune response. GBP1 expression was increased in JEV-infected HeLa cells and a colocalization with JEV-NS1 as part of the replication complex was identified. Upon IFNγ-treatment, silencing of GBP1 resulted in decreased viral titers and in the activation of the JAK-STAT signaling via phosphorylation of STAT1 [29].

In addition, murine GBP4 and hGBP7 have been identified to behave pro-virally. This is achieved by interference with factors involved in the antiviral innate immunity. GBP4 has been reported to interact with IFN-I regulator factor 7 (IRF7), an essential regulator involved in type I IFN expression. Thereby, it disrupts the IRF7-TRAF6 interaction, resulting in an impaired IFNα production [30]. A previous study by Feng et al. [31] described GBP7 as facilitating factor in IAV replication through interference with the NF-κB and JAK-STAT signaling cascades. This went along with the expression of type I and type III IFNs and proinflammatory cytokines.

Within the dynamin superfamily, GBPs share high homology with atlastins. In line with pro-viral effects ascribed to GBPs, ER-resident atlastins, which are involved in fusion and tethering of ER tubules [54], have been identified to promote replication and assembly of dengue (DENV) and Zika virus (ZIKV) [55]. A previous study identified GBP2/5, two GBP1-prenylation paralogs, to be upregulated upon ZIKV infection, thus leading to reduced viral replication through inhibition of the host protease furin [56]. In line with this, we observe an increased GBP1 expression and protein amount in ZIKV-infected cell. However, due to the GBP2/5-mediated antiviral effect, we assume that in context of ZIKV-infected cells, GPB1 acts as an antiviral host-factor.

Under physiologic conditions, GBPs are expressed in various cell types with GBP1 displaying a granular or diffuse staining pattern in the cytoplasm of the cells [20]. Upon IFNγ-

treatment, GBP1 partially colocalizes with components of the phagolysosomal system such as early- or late endosomes and lysosomes in a granular or vesicular shape [57]. In Huh7.5 cells, GBP1 is expressed at a very low basal level. Elevated levels of GBP1 after induction by IFNγ or due to overexpression, result in a vesicular GBP1 distribution in the perinuclear region in the HCV infected cells. This could not be observed in HCV negative cells, where GBP1 is equally distributed in the cytoplasm. Remarkably, we found GBP1 colocalizing with HCV core around LDs. The LD targeting of GBP1 requires the presence of HCV core, as we were not able to detect GPB1 around LDs in the subgenomic replicon cells Huh9-13 that lack the HCV structural proteins. However, after overexpression of HCV core in the Huh9-13 cells we found GBP1 around LDs. This obvious difference between cells replicating intact HCV genomes and cells replicating subgenomic replicons [27] could be causative for conflicting data describing the HCV-GBP1-crosstalk. However, not all LDs were coated by GBP1 and HCV core. This is in line with previous studies that describe only 20% of the LDs in the Jc1-infected cells colocalizing with HCV core [52].

Infection with HCV leads to a massive reorganization of intracellular membranes and organelles to form the membranous web. This consists of LDs and DMVs in close proximity to the ER. The DMVs are involved in HCV replication, whereas the LDs represent platforms being essential for the viral assembly. Based on the observation that GBP1 and HCV core colocalize on the surface of LDs, GBP1 might be part of an assembly platform involved in the morphogenesis and release of viral particles. Indeed, silencing of GBP1 resulted in a decrease of intra- and extracellular viral titers and in an intracellular accumulation of the core structures. These observations corroborate the protein's role during the release of viral particles. Moreover, silencing of GBP1 leads to a decreased specific infectivity (number of infectious viral particles/genomes), reflecting the point that viral morphogenesis is disturbed. Ultimately, this results in an increased number of defective or improperly assembled particles. In addition, NS3 protein levels were significantly reduced, reflecting a decrease in viral replication. This effect was even more pronounced after IFNγ stimulation, indicating that GBP1 is not required for the antiviral capacity of IFNγ.

However, coimmunoprecipitation experiments, which were aimed to corroborate the super resolution microscopy-based data revealing a tight colocalization of GBP1 and core, failed. An explanation could be that additional factors at the membrane environment are required to mediate the association between GBP1 and core. Furthermore, it can be speculated that the autophagy adaptor protein p62 might be involved in this process. p62 interacts with ubiquitinated proteins via its ubiquitin-associated (UBA) domain and delivers its cargo towards autophagosomes via its LC3 interacting region (LIR) or to the proteasome via its N-terminal Phox-BEM1 (PB1) domain [58]. For HCV, p62 has been identified to be essential for the formation of autophagosomes [11]. In line with this, immunofluorescence analyses of this study indicate a colocalization of the three proteins (GBP1, HCV core, p62) (**S4A and S4B Fig**).

The formation of the membranous web (MW) enables the spatiotemporal separation of viral translation, replication and assembly and shields the replication organelles (ROs) from the host innate immune machinery [59]. Hence, the access of proteins to this network is tightly controlled by components of the nuclear import machinery [60]. The non-prenylated GBP1 paralogs GBP3 and GBP4 are able to localize to the nucleus, however, this has, so far, not been described for GBP1 [21]. In accordance to this, based on prediction tools, GBP1 does not harbor a classical nuclear localization signal (NLS). Thus, the access of GBP1 to the MW must be mediated by a different mechanism. Recent studies indicate a direct binding of GBP1 to phosphatidylinositol-4-phosphate (PI4P) upon endolysosomal damage [61]. In HCV infected cells, PI4P accumulates in the MW to enrich the ER-derived membranes in cholesterol by an OSBP-mediated PI4P-cholesterol exchange [62]. Based on this, we hypothesize that PI4P might

facilitate the access of GBP1 to the MW. In addition, HCV core has been described to contain putative NLS sequences [63]. Additionally, it has been previously reported that HCV core recruits the nuclear pore protein, a component of the nuclear pore complex, to the assembly platform, where it binds the viral RNA and favors viral assembly [64].

Along with the role of p62 for autophagosome-formation, actin-remodelling has gained attention in the context of phagophore morphogenesis. It has been previously reported, that GBP1 interacts with the actin cytoskeleton and IFNγ-dependent GBP1 induction triggers the reorganization of actin filaments [46]. In accordance to this, GBP1 interferes with the nuclear transport of KSHV particles and pseudorabies virus by disrupting actin filaments [23,24]. Indeed, in the HCV-positve cells, we found GBP1 and HCV core colocalizig with actin filaments. However, we did not detect any effect on the actin filaments upon IFNγ-treatment in the HCV-positve or HCV-negative cells. One possible explanation could be the use of different cell culture systems. Further, actin is involved in the shaping of autophagosomes [65]. Here, polymerization of actin facilitates the formation of highly-curved membranes and phago-phores. Accumulation of the actin network finally helps to shape the autophagosome. HCV particle release is tightly connected to the autophagosomal pathway [66,67]. Hence, as we see GBP1 and HCV colocalizing with actin filaments, it is tempting to speculate whether the GBP1-actin interaction further facilitates HCV viral particles release.

Moreover, when analyzing the lysosomal system, we found that silencing of GBP1 interferes with the morphology of LAMP2 structures and the lysosomal acidification. LAMP2 is an essential factor involved in maturation of autophagosomes. In line with this, LAMP2-deficient hepatocytes show an impaired autophagosomal maturation [68]. It has been described that HCV hijacks the autophagosomal pathway for efficient release of viral particles and autophagy in turn is involved in several steps of the HCV life cycle [69,66]. Thus, the autophagosome represents a branching point during HCV morphogenesis. On the one hand, a minor fraction of *de novo* synthesized HCV particles is released via amphisomes and MVBs. On the other hand, the major fraction ends up in the autolysosomal compartment and is degraded. In light of the GBP1-dependent impact on lysosomal maturation, we conclude that silencing of GBP1 interferes with formation of autophagosomes, thus preventing this essential part in release of viral particles. This is corroborated by the decrease of intra- and extracellular viral titers and the observation that HCV core accumulates outside LAMP2 positive structures. Due to the impaired lysosomal acidification, the excess of core proteins cannot be degraded, which then accumulate inside the cell. In line with this, it can be speculated that these core-associated structures represent misfolded protein aggregates. Interestingly, we observe an increased LAMP2 signal and lysosomal acidification in HCV infected cells after IFNγ treatment. This stands in contrast to the literature, which reports an impairment in autophagy and LAMP1/2 signal after IFNγ-treatment in A549 cells [70]. Further experiments will be required to determine the link between IFNγ-dependent GBP1 expression and LAMP2 maturation that is essential for the HCV morphogenesis. To this end, induction of autophagy by treatment with rapamycin could restore the siGBP1-mediated effect of the impaired lysosomal acidification and further increased the HCV core levels. However, the effect of Rapamycin treatment seems to be independent from GBP1 and needs to be further analyzed. For HEV, GBP1 has been described to target capsids towards lysosomal degradation [25] and recent studies of our group identified HCV core partially localizing within lysosomal structures [47,11]. In the present study, we detected HCV core within LAMP2-positive structures. However, the targeting of HCV core towards lysosomal degradation was independent of GBP1, thus supporting our observations of a different function of GBP1 for the HCV life cycle. So far, degradation of HCV core has been linked to the ubiquitin-proteasome pathway through core ubiquitination by the E3 ligase E6-associated protein (E6AP) [71,72]. Alternatively, a p62-dependent selective

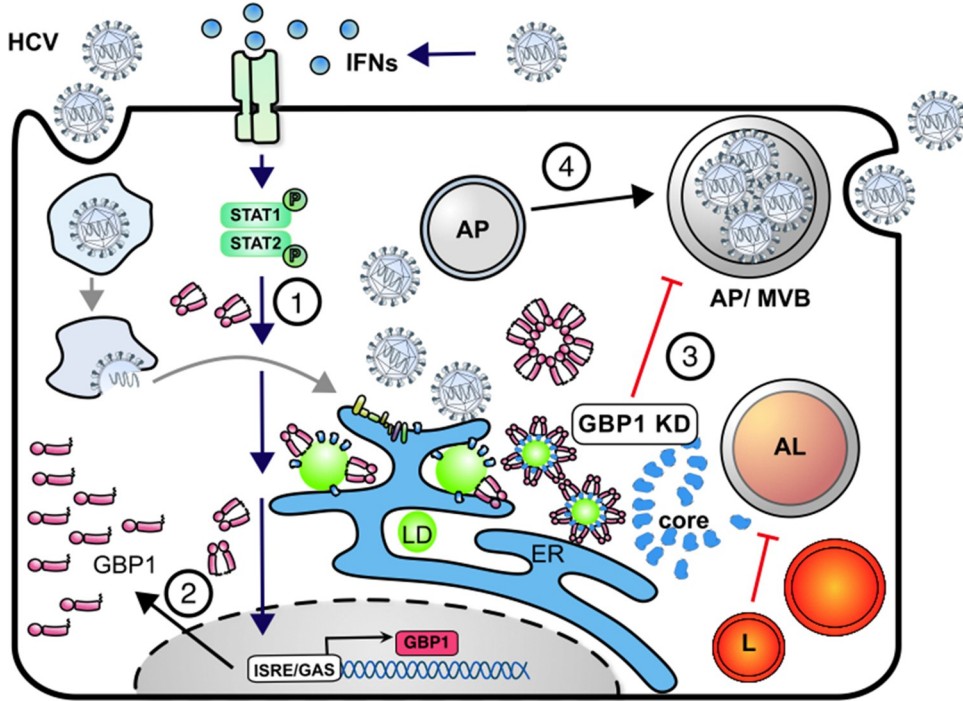

**Fig 8. Proposed model for the pro-viral role of GPB1 during the HCV morphogenesis and assembly. (1)** Infection with HCV activates the host innate antiviral response resulting in the induction of IFNs and interferon stimulated genes. **(2)** In line with this, we observe that the expression of the ISG GBP1 is upregulated upon HCV infection *in vivo* and *in vitro*. **(3)** Silencing of GBP1 impaired the release of viral particles and resulted in intracellular HCV core accumulation and impaired lysosomal acidification. **(4)** GBP1 overexpression resulted in increased viral release, and a vesicular distribution of GBP1 in the perinuclear region colocalizing with HCV core around LDs as part of the assembly platform, reflecting an impact on viral morphogenesis and release. AL, autolysosome; AP, autophagosome; ER, endoplasmic reticulum; GAS, interferon-gamma activated sequence; GBP1, guanylate binding protein 1; IFN, interferon; ISRE, Interferon stimulated response element; MVB, multivesicular bodies.

degradation upon induction of oxidative stress, independent from the ubiquitin-proteasome system, has been reported [73].This supports our hypothesis of p62 acting as adapter protein linking HCV core and GBP1 to autophagosomes for efficient release.

In conclusion, our study identified a new pro-viral role for GBP1 in the HCV life cycle. This stands in contrast to its currently described antiviral capacity. We demonstrate that GPB1 is an essential factor involved in viral assembly and release. GBP1 colocalizes together with the HCV core protein on the surface of LDs. Here, it is part of the assembly platform and is involved in the targeting of viral particles towards the autophagy-associated vesicles for efficient release of viral particles (**Fig 8**).

## Supporting information

**S1 Fig. Efficient IFN-dependent GBP1 induction in Huh7 cells.** Luciferase reporter gene assay of Huh7 cells transfected with pGL3-GBP1-Luc plasmid DNA (GBP1-Luc) to monitor GBP1 expression after treatment with different concentrations (100 U/ml, 250 U/ml, 500 U/ml) of IFNα, IFNβ, IFNγ for 24 h (48 h post seeding). Relative values are referred to the untreated control (UT) (set to 1). N = 9 technical replicates.
(TIF)

**S2 Fig. GBP1 can be silenced by selected siRNA and is not compensated by GBP2 or GBP5.** **(A)** qPCR analysis to monitor GBP1 mRNA levels 72 h after GBP1 silencing in stable HCV positive Huh7.5 cells (Jc1). Graph shows average percent change of the GBP1 levels referred to the untreated (UT) scrambled (scr RNA) control (set to 100). N = 5 biological replicates. Statistics were performed as mean ± SEM, unpaired t-test referred to ctrl. with $^{**}$p<0.01, $^{****}$p<0.0001. **(B, C)** qPCR analysis to monitor GBP2 (B) and GBP5 (C) mRNA levels 72 h after GBP1 silencing in stable HCV positive cells (Jc1). Relative change of GBP2/5 levels were referred to the untreated (UT) scrambled (scr RNA) control (set to 1). N = 5 biological replicates. Statistics were performed as mean ± SEM, unpaired t-test referred to ctrl. with $^{**}$p<0.01, $^{****}$p<0.0001.
(TIF)

**S3 Fig. Lysosomal targeting of HCV core does not depend on.** CLSM analysis of stably HCV replicating Huh7.5 cells transfected with a GBP1-Wt encoding construct using GBP1- (red), HCV core (green)-, LAMP2 (cyan) and p62 (magenta)-specific antisera. To inhibit lysosomal degradation the cells were treated with Leupeptin (200μM) for 24 h or left untreated as a control (UT). The cells were analyzed 48 h pt. Scale bars, 20 μm.
(TIF)

**S4 Fig. p62 colocalizes with GBP1 and HCV core positive structures. (A)** CLSM analysis and 3D reconstructions (3D inset) of stable HCV positive cells (Jc1) to visualize p62 colocalizing with GBP1 and HCV core. 24 h post seeding the cells were transfected with a GBP1-Wt construct for 48 h and fixed with a 1:1 mixture of ice-cold ethanol/acetone for 10 min. The cells were stained for GBP1 (red), core (green), and p62 (cyan) using specific antisera. Nuclei were visualized with DAPI. Scale bar, 10 μm. Images are representative of 3 biological replicates. **(B)** CLSM analysis, z-stacks and 3D reconstructions (3D inset, 3D xy-sections) of stable HCV positive Huh7.5 cells (Jc1) to visualize p62 colocalizing with GBP1 and HCV core. To induce GBP1 expression the cells were treated with IFNγ (100 U/ml) 24 h post seeding. After 24 h the cells were fixed with a 1:1 mixture of ice-cold ethanol/acetone for 10 min. The cells were stained for GBP1 (red), core (green), and p62 (cyan) using specific antisera. Nuclei were visualized with DAPI. Scale bar, 10 μm. Images are representative of 3 biological replicates. All immunofluorescence images were deconvoluted using the LasX Lightning Tool using the adaptive algorithm.
(TIF)

## Acknowledgments

The authors thank Kerstin Schwickert, Marie-Luise Theuerkauf, and Patrycja Dudek for their topic-related discussions.

## Author Contributions

**Conceptualization:** Daniela Bender, Gerrit J. K. Praefcke, Eberhard Hildt.

**Formal analysis:** Daniela Bender.

**Funding acquisition:** Eberhard Hildt.

**Investigation:** Daniela Bender, Alexandra Koulouri, Xingjian Wen, Mirco Glitscher, Anja Schollmeier, Liliana Fernandes da Costa, Robin Oliver Murra, Gert Paul Carra, Vanessa Haberger.

**Methodology:** Daniela Bender, Eberhard Hildt.

**Project administration:** Daniela Bender, Eberhard Hildt.

**Resources:** Daniela Bender, Eberhard Hildt.

**Supervision:** Daniela Bender, Eberhard Hildt.

**Visualization:** Daniela Bender, Mirco Glitscher.

**Writing – original draft:** Daniela Bender.

**Writing – review & editing:** Mirco Glitscher, Gerrit J. K. Praefcke, Eberhard Hildt.

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
