## [Decision Letter · Decision Letter 0]

18 Oct 2023

Dear Prof. Dr. Hildt,

Thank you very much for submitting your manuscript "Guanylate-Binding Protein 1 acts as a pro-viral factor for the life cycle of  Hepatitis C virus" for consideration at PLOS Pathogens. As with all papers reviewed by the journal, your manuscript was reviewed by members of the editorial board and by several independent reviewers. In light of the reviews (below this email), we would like to invite the resubmission of a significantly-revised version that takes into account the reviewers' comments.

The reviewers appreciated the novelty of the results demonstrating the role of GBP1 in the HCV virion assembly but raised concerns about insufficient controls in some experiments and overinterpretation of the results obtained in Huh7.5 cell line defective in IFN signaling.  We agree that it is important to validate key findings using different virus-cell culture systems. 

We cannot make any decision about publication until we have seen the revised manuscript and your response to the reviewers' comments. Your revised manuscript is also likely to be sent to reviewers for further evaluation.

Sincerely,

George A. Belov, PhD

Academic Editor

PLOS Pathogens

Alexander Gorbalenya

Section Editor

PLOS Pathogens

Kasturi Haldar

Editor-in-Chief

PLOS Pathogens

orcid.org/0000-0001-5065-158X

Michael Malim

Editor-in-Chief

PLOS Pathogens

orcid.org/0000-0002-7699-2064

Reviewer's Responses to Questions

**Part I - Summary**

Reviewer #1: This paper presents a detailed cell biological analysis of the role of the cellular protein GBP1 in the HCV lifecycle. The experiments are carefully performed and presented clearly. The data demonstrate convincingly that GBP1 colocalises with HCV Core and lipid droplets and the authors speculate that GBP1 may play a role in virion morphogenesis and release. I have only minor comments on the details of the paper (see following sections).

My main concern is the overall interpretation and significance of the study. Despite the convincing data about colocalisation of GBP1 with Core/LD, the effects of GBP1 silencing are modest at best. Fig 3B shows no effect on HCV Core protein, Fig 3G and 4E no effect on intracellular or extracellular viral RNA, and Fig 4D only a 2-fold effect on virus titre. Thus it is disingenuous to state that GBP1 is 'essential' for virus assembly.

Reviewer #2: The study of Daniela Bender and co-workers aims to study the role of the human guanylate binding protein 1 (GBP1) for the replication and egress of hepatitis C virus (HCV) particles. As GBP1 is an interferon-response gene exhibiting both anti-viral and pro-viral effects on serval viruses, the authors studied its impact on HCV using HCV replicon cells using RNAi, confocal and high-resolution microscopy, qPCR, Western blotting, and luciferase reporter assays. The authors revealed that HCV induces GBP1 expression, which then colocalizes with HCV core protein around lipid droplets at the replication complex and promote viral assembly. Moreover, the authors suggest that GBP1 associates with the autophagosomes regulating their function and maintaining the specific infectivity of HCV viral particles.

The manuscript is well written, the results are very well represented and highly relevant for the understanding of virus-induced evasion of the interferon response and interesting. In particular, the findings contribute to the understanding of the HCV life cycle by exploiting antiviral host factors like GBP1. However, to strengthen the conclusions, additional controls must be included, and key findings reproduced using an additional HCV infection model. Moreover, as GBP1 has been previously suggested to mediate interferon-induced actin reorganization, appropriate co-staining of actin need to be included in key experiments to control potential indirect effects of GBP1 on lysosomal and autophagosomal function via actin filament reorganization after perturbation studies (see specific comments).

Reviewer #3: Authors of this manuscript tried to decipher the role of GBP1 during HCV life cycle. GBP1 belongs to IFN inducible subfamily of guanosine triphosphatases (GTPases) that have been reported to exert broad anti-microbial activity and regulate host defenses against several intracellular pathogens. Enhanced GBP1 expression has been reported both at transcript and protein levels upon some RNA virus infection, which also described GBP1 association with the virus replication membranes.

It was reported earlier that GBP-1 inhibits HCV subgenomic replication. HCV replication and virus production were suppressed significantly by overexpression of GBP-1, along with other ISGs, IFI-6-16, or IFI-27. Interestingly, HCV-NS5B protein was shown to bind GBP-1 and countered the antiviral effect by inhibition of its GTPase activity.

However, the results of the current study are in contrast to previous observations. Here, authors have demonstrated the role of GBP1 as an essential factor involved in viral assembly and release. It appears GBP1 protein colocalizes with the HCV core protein on the surface of lipid droplets (LDs) and targets the viral particles towards the autophagy-associated vesicles for efficient viral release from infected cells.

Earlier studies have shown decreases in GBP1 expression during HCV infection; however, this study shows increase in GBP1 infection during HCV infection. These contrasting observations could be due to the choice of different cell lines. Huh7.5 may not be ideal to study interferon response.

**Part II – Major Issues: Key Experiments Required for Acceptance**

Reviewer #1: Generally there is a lack of controls - for example in Fig 1 is there a control to show that GBP1 is specifically induced by HCV infection?

It is hard to equate the different effects of HCV on GBP1 expression - in Fig 1 HCV increases GBP1 but in Fig 2 it reduces it. The converse is also true - the effect of GBP1 on HCV is discrepant. eg GBP1 silencing has no effect on Core or RNA but decreases NS3 expression (Fig 3E).

I would like the see the absolute values for virus titre and RNA levels in Fig 4. Fold changes may hide interesting data.

Reviewer #2: (Major)

1. It has been previously reported that GBP1 binds to actin and influence actin filament reorganization (e.g., PMID 24190970). As actin has a role in the shaping of autophagosomes (PMID 27723370), this must be controlled by co-staining of actin in some of the experiments of Fig.3,4,5 and discussed.

2. Key experiments in Fig. 1 and Fig. 3F must be validated using infectious HCV in cell culture. This would allow comparing the effect in uninfected and infected cells side by side.

3. Fig. 1B, 2D: an additional virus known to induced GBP1 should be shown as positive control. Ideally a virus, that infects the same cell line, e.g., VSV or HEV.

4. Fig.1D: Was only one patient being analysed? The number of patients, slides and cells analyses must be clearly described in the figure legend.

5. Fig. 3F: The number of data points in the CTCF graph and the statement in the legend >3 biological replicates + the number of cells per replicate analyses do not match and require clarification.

6. Fig. 4C: Statistics must be provided for the quantification of lysotracker/LAMP2 including error bars.

7. What is the effect of inhibitors / activators of autophagy on the GBP1 LAMP2 axis? Can an autophagy inducer like inhibitors of Akt or mTOR rescue the effect observed in Fig. 4C?

8. A positive control validating leupeptin function in this experimental setting must be shown.

9. Fig. 6A: The NS3 level in the representative Western blot is not reflected by the respective quantification in the bar graph. Is the NS3 “increase” in the Western blot rather due to changes of actin expression (see major comment 1)? An alternative control instead actin is required.

10. The authors demonstrate a co-localization of core and GBP1 around the lipid droplets of in the membranous web. It has been previously demonstrated that HCV restricts access to the membranous web through nuclear pore-like structures and nuclear import machinery (PMID 26863439). Does GBP1 possess a nuclear localization signal? It seems to be able to shuttle into the nucleus based on prediction tool. This should be discussed.

Reviewer #3: The following comments might help the authors to improve the manuscript.

1. The HCV-JFHI infection cell culture experiments are carried out in Huh7.5, which are RIG-I mutated and lack type-I-IFN response. It is important to use HepG2 cells or Huh7 cells (use HCV-H77S virus) for better clarity.

2. All experiments in Fig 1 are done in JC1 stable expressing cell line. It is important to investigate the protein levels of GBP1 by western blotting upon HCV infection in an uninfected cell line.

3. Also, the time kinetics of GBP1 expression during HCV infection in above cell types (with JFHI, JCI and H77S viruses) will be informative for better correlation.

4. In Fig 1D, the liver sections from HCV infected patients show an increase in GBP1 expression. Authors need to mention the HCV genotype of patients used in the experiments and co-relate with their experiments.

5. In Fig 2A, the independent biological replicates should be at least n=3 (for IFN-α & IFN-β only n=2). It is necessary to analyze the experiments at same biological replicates.

6. For Fig 2 B to 2D, the relative difference between Untreated cells with GND or JC1 also needs to be depicted in the graph to analyse the effect of only JC1. Possibly the basal levels of GBP1 are higher in JC1 cell line and therefore the further increase upon IFN induction is lesser. These experiments can also be repeated with virus infection in place of the use of stable cell lines.

7. Fig 2A-C, upon IFN-γ treatment under GND conditions, there is more GBP1 promoter activity, mRNA level (even though they are not significant). It seems that IFN-γ + GND together increases GBP1 expression. Need to clarify.

8. Fig 2D and 3A (scr RNA-GND), the expression of GBP1 is nearer (prominent) to the peri-nuclear space while this effect is less (diffused) during HCV-JC1 infection condition. Also, not all cells show increased expression of GBP1 during JC1 expression (core). This can be clearly seen in the merged section as well. Results are confusing!

9. In Fig 3A, under GND UT conditions, the GBP1 expression is defined around the peri-nuclear space while under Jc1 UT conditions, the expression of GBP1 is more diffused- the change in the expression of GBP1 is expected? It would be better to show GBP1 expression under GND-UT conditions compared with JC-1 UT conditions.

10. In Fig 3B, the densitometry does not match with the representative western blots. The expression of GBP1 under siGBP1-IFN-γ is shown less in densitometry than siGBP1-UT conditions. Re-analysis of the western blot data is required.

11. In Fig 3D, quantification of only 4 cells might provide an erratic view. This number needs to be increased to justify the observation.

12. In Fig 3E, instead of using unpaired t-test, it would be better to use one-way ANOVA analysis between the scRNA and siGBP1 (UT and IFN-γ). Is there any role of GBP1 upon HCV replication, since the authors have shown the expression of NS3 decreasing under si-GBP1 (IFN-γ) condition?

13. Fig 4B, authors have shown that upon siGBP1, core co-localize with LAMP2, what happens to NS3 in the same condition? Again, in Fig4C-D, the analysis used is un-paired t-test, instead one-way ANOVA should be used.

14. Fig 4F, could this decrease in specific infectivity be linked to the decrease in Core protein levels and inefficient packaging?

15. Fig 5B, the LAMP structures mentioned previously in Fig 4B to be induced upon siGBP1 are now visible upon IFN treatment which increase GBP1. Need to elaborate on this.

16. In Fig 6, the authors have shown that upon GBP1 over-expression, HCV-core protein co-localizes with GBP1 at peri-nuclear space, what happens during siRNA conditions?

17. Fig 6. What happens to extracellular RNA levels and intracellular RNA titres upon GBP1 overexpression? This is essential to be compared with fig 4 D-F data to define an inverse correlation upon silencing or overexpression of GBP1 and to specify the role of GBP1 on packaging or release of the virus particles.

18. Fig 6E. and 3C. Both silencing and overexpression of GBP1 leads to perinuclear core aggregation. Not clear why?

19. Fig 7F, the observation is very interesting. However, it is intriguing why such structures of GBP1 were not visible in intracellular GBP1 staining in the previous experiments and also in the experiment 7G? It would be important to quantify the percentage of cells exhibiting such staining in this context.

20. In Fig 7G, authors have used Huh9-13 replicon model for showing the co-localization of GBP1, dsRNA and LDs. A better experiment would be, over-expression of core protein in Huh9-13 cells and then see if the association still exists.

21. Fig 8, the graphical abstract can be made clearer by proper labelling the pathway as a, b, c, etc.

**Part III – Minor Issues: Editorial and Data Presentation Modifications**

Reviewer #1: n/a

Reviewer #2: 1. Fig. 6E: the magnification of the additional zoom window within the inset should be stated in the legends.

2. Material and Methods: the use of upper and lower cases in the subtitles.

3. Typo line 230: “4°C.bHorse..”

4. The order appearance of Figure panels should be respected: Fig. 4C cited before Fig. 4B

Reviewer #3: 1. As a comparison of Fig 1 B and 1 C, the localisation of GBP1 seems to be inconclusive. In 1C, it seems cytoplasmic but in 1B, the staining is very faint and not cytoplasmic. Please comment on this or provide a better representative image.

2. The manuscript requires extensive English and grammatical corrections.

3. Difference in referencing style in discussion section, line no: 473-474

4. Grammatically wrong sentence, line 519-521.

5. In the material and method section, several sentences are abruptly ending, (for e.g., line 89-90). Also, figure legend 5, sentence is abruptly ending.

PLOS authors have the option to publish the peer review history of their article (what does this mean?). If published, this will include your full peer review and any attached files.

Reviewer #1: No

Reviewer #2: No

Reviewer #3: No
---

## [Decision Letter · Decision Letter 1]

16 Jan 2024

Dear Prof. Dr. Hildt,

We are pleased to inform you that your manuscript 'Guanylate-Binding Protein 1 acts as a pro-viral factor for the life cycle of  Hepatitis C virus' has been provisionally accepted for publication in PLOS Pathogens.

Best regards,

George A. Belov, PhD

Academic Editor

PLOS Pathogens

Alexander Gorbalenya

Section Editor

PLOS Pathogens

Michael Malim

Editor-in-Chief

PLOS Pathogens

orcid.org/0000-0002-7699-2064

Reviewer Comments (if any, and for reference):

Reviewer's Responses to Questions

**Part I - Summary**

Reviewer #2: The study of Daniela Bender and co-workers aims to study the role of the human guanylate binding protein 1 (GBP1) for the replication and egress of hepatitis C virus (HCV) particles. As GBP1 is an interferon-response gene exhibiting both anti-viral and pro-viral effects on serval viruses, the authors studied its impact on HCV using HCV replicon cells using RNAi, confocal and high-resolution microscopy, qPCR, Western blotting, and luciferase reporter assays. The authors revealed that HCV induces GBP1 expression, which then colocalizes with HCV core protein around lipid droplets at the replication complex and promote viral assembly. Moreover, the authors suggest that GBP1 associates with the autophagosomes regulating their function and maintaining the specific infectivity of HCV viral particles.

Reviewer #3: The authors have addressed all the queries appropriately.

**Part II – Major Issues: Key Experiments Required for Acceptance**

Reviewer #2: (No Response)

Reviewer #3: (No Response)

**Part III – Minor Issues: Editorial and Data Presentation Modifications**

Reviewer #2: (No Response)

Reviewer #3: (No Response)

PLOS authors have the option to publish the peer review history of their article (what does this mean?). If published, this will include your full peer review and any attached files.

Reviewer #2: No

Reviewer #3: No

---

## [Editor Report · Acceptance letter]

31 Jan 2024

Dear Prof. Dr. Hildt,

We are delighted to inform you that your manuscript, "Guanylate-Binding Protein 1 acts as a pro-viral factor for the life cycle of  Hepatitis C virus," has been formally accepted for publication in PLOS Pathogens.

Best regards,

Michael Malim

Editor-in-Chief

PLOS Pathogens

orcid.org/0000-0002-7699-2064